# Procedural Pretraining: Warming Up Language Models with Abstract Data

**Liangze Jiang** [* 1 2]   **Zachary Shinnick** [* 3]   **Anton van den Hengel** [3]   **Hemanth Saratchandran** [3]   **Damien Teney** [2]

## Abstract

Pretraining language models directly on web-scale corpora is the de facto paradigm. We study an alternative where the model is initially exposed to *abstract structured data* to ease the subsequent acquisition of rich semantic knowledge, much like humans learning simple logic and mathematics before higher reasoning. We focus on *procedural data*, generated by formal languages and other simple algorithms, as such abstract data.

We first diagnose the algorithmic skills that different forms of procedural data can improve, often significantly. For example, the accuracy of context recall (NEEDLE-IN-A-HAYSTACK) jumps from 10 to 98% when a model is pretrained on Dyck sequences (balanced brackets). Second, we study how these gains are reflected in pretraining larger models (up to 1.3B). We find that front-loading as little as 0.1–0.3% procedural data significantly outperforms standard pretraining on natural language, code, and informal mathematics (C4, CODEPARROT, and DEEPMIND-MATH datasets). Notably, this also enables the models to reach the same loss value with only 55/67/86% of the original data and thus a comparable reduction in FLOPs. Third, we explore the mechanisms behind the benefits and find that procedural pretraining instils non-trivial structure in both attention and MLP layers. The former is particularly important for structured domains (e.g. code), and the latter for language. Finally, we lay a path for combining multiple forms of procedural data. Our results show that procedural pretraining is a simple, lightweight means to improving performance and accelerating language model pretraining, ultimately suggesting the promise of disentangling knowledge acquisition from reasoning in LLMs. Code is available at this page.

*Equal contribution [1]EPFL [2]Idiap Research Institute [3]AIML, Adelaide University. Correspondence to: LJ <liangze.jiang@epfl.ch>, ZS <zachary.shinnick@adelaide.edu.au>.

*Proceedings of the 43$^{rd}$ International Conference on Machine Learning*, Seoul, South Korea. PMLR 306, 2026. Copyright 2026 by the author(s).

## 1. Introduction

Large language models (LLMs) simultaneously acquire multiple forms of knowledge during pretraining. They absorb rich semantic content, but also acquire abilities for manipulating this knowledge. This entangled learning of knowledge and abstract skills has been identified as a key limitation of current models (Han et al., 2025; Kumar et al., 2025), leading to their reliance on surface-level heuristics rather than systematic reasoning procedures (Nikankin et al., 2025).

To mitigate knowledge-reasoning entanglement, we study *using abstract, structured data to 'warm up' language models*. Intuitively, this is a lightweight pretraining that builds algorithmic scaffolding without relying on semantic shortcuts, much like infants learning games like stacking blocks (Smith & Gasser, 2005) before moving to sophisticated reasoning and knowledge. With **procedural pretraining**, we posit that early exposure of LMs to procedural data[1] facilitates and enhances standard pretraining on semantically-rich corpora.

In prior work, Hu et al. (2025) showed that "pre-pretraining" LLMs on data generated from formal languages yields more value per token than natural language. Wu et al. (2022) and Zhang et al. (2024) successfully used data from simple algorithms and cellular automata. Their findings echo the established practice of pretraining on computer code, another structured domain thought to aid learning compositional and recursive reasoning (Petty et al., 2024). Prior works, however, typically treat procedural data as either *imitation* of linguistic properties, or as a drop-in *substitute* of standard pretraining. In contrast, we study procedural data from a broader *algorithmic* view, and position it explicitly as a *complementary* warming-up stage for standard pretraining. Our experiments contain two main parts. The **first part** identifies why and when procedural pretraining helps using algorithmic tasks as diagnostic tools, while the **second part** shows, with larger models, procedural pretraining improves standard pretraining in several domains of practical interest across scales. Our contributions are:

**(1) Probing procedural pretraining with algorithmic tasks.** We find that different forms of procedural data

---

[1]We use procedural data to refer to the output of explicit algorithms (e.g. formal languages or sorting), in contrast to synthetic data, which is typically generated by trained models such as LLMs.

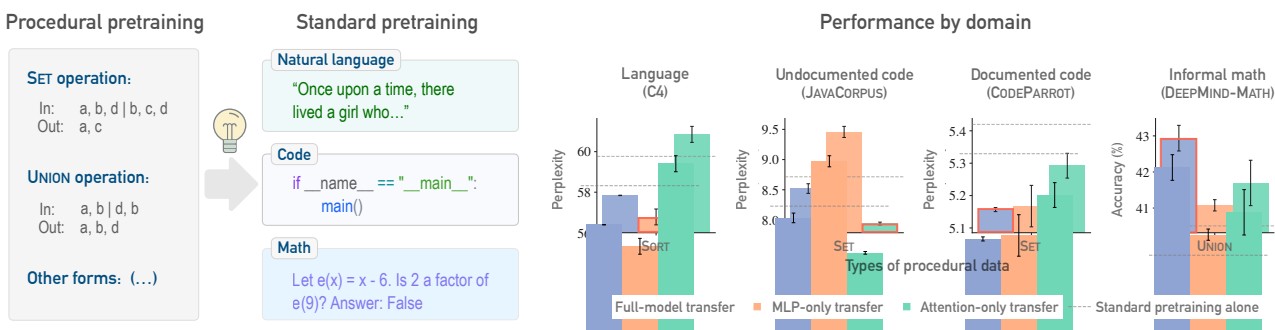

*Figure 1.* (**Left**) We pretrain language models on procedural data before exposing them to standard datasets of language, code, or mathematics. The procedural data is generated with simple algorithms and aims to teach elementary skills to aid the acquisition of semantic knowledge. (**Right**) This lightweight initial step speeds up standard pretraining and improves performance on diverse domains, with different pretrained layers (MLP vs. attention) contributing differently to each domain.

each enhance specific algorithmic skills (Section 4.1). The pretrained information also proves to be localised in specific layers (attention vs. MLPs, Section 4.2). We also rule out simplistic explanations that could account for the observed improvements, such as rescaling the initialisation or a generic attention sharpening (Section 4.3).

**(2) Transfer to pretraining on diverse domains.** We show that the improvements on algorithmic skills transfer to multiple semantic domains, namely natural language, code, and informal mathematics (Section 5.1–5.2). The information learned in procedural pretraining proves to be *complementary* to standard pretraining datasets. For example, we consistently improve over standard pretraining with as little as $0.1 - 0.3\%$ extra procedural tokens. Procedural data also proves to be an *efficient substitute* to standard data. On C4, CODEPARROT, and DEEPMIND-MATH datasets, it enables models to reach the same loss with respectively 55%, 67%, and 86% of the original data and therefore a comparable FLOP reduction. Furthermore, we validate these findings across different model sizes (up to 1.3B parameters) and data sizes (up to 10.5B tokens), and show that the gains at standard pretraining persist on downstream language, code generation and commonsense reasoning tasks (Section 5.3).

**(3) Localising transferable pretrained information** (Section 5.4). We localise useful procedurally-pretrained information for each domain. We find that the attention layers are more important for structured, language-free domains like pure code, while MLP layers help natural language more. The latter is intriguing because MLPs are believed to store factual knowledge in LLMs (Dong et al., 2025; Geva et al., 2020; Xu & Chen, 2025) which procedural data cannot directly provide. On data containing both natural language and structured data such as CODEPARROT (*documented* code) and DEEPMIND-MATHEMATICS (*informal* mathematics), both components are important (Figure 1 right).

**(4) Combining the benefits of different forms of procedural data** (Section 6). We explore two techniques and obtain

promising results by either pretraining on a mixture of data types, or surgically combining weights of several pretrained models. This lays out several directions for future work.

Our results show that procedural data is both a data-efficient alternative and an effective complement to standard pretraining. We discuss intriguing future directions in Section 7 and how our findings may ultimately improve the knowledge and reasoning acquisition in LLMs.

## 2. Related Work

The linguistic literature contains a number of results on training language models with artificial data. These works often use formal languages to imitate properties of natural language (Chiang & Lee, 2022; Goodale et al., 2025; McCoy & Griffiths, 2023; Papadimitriou & Jurafsky, 2023; Ri & Tsuruoka, 2022; Hu et al., 2025). In contrast, we follow a more general algorithmic perspective, and find how different types of procedural data can improve specific algorithmic skills. Hence, we also study benefits on domains beyond language, namely code and informal mathematics.

We use procedural data generated with simple algorithms and cellular automata, as in recent works (Lindemann et al., 2024; Wu et al., 2022; 2021; Zhang et al., 2024; Bloem, 2025). These works focus on procedural data as a *substitute* for standard pretraining data. In contrast, we also evaluate procedural data as a *complement*, and find that it can impart capabilities lacking from standard semantic data across diverse domains. Additionally, we validate empirically that the benefits of procedural pretraining even persist after further fine-tuning on downstream tasks. We also analyse in greater depth the mechanisms behind the empirical benefits, such as the localisation of pretrained knowledge in MLP vs. attention layers. This sometimes also reveals further empirical pretraining gains by only transferring specific layers from procedural pretraining. Finally, while most of this existing work focuses on a single type of data, we take steps

towards combining multiple types of procedural data, which lays out a path for important next steps for this line of work.

A concurrent work (Shinnick et al., 2026) shows that procedural data benefits visual learning. Together with our findings, this implies that procedural data might inject modality-agnostic mechanisms (Huh et al., 2024) into the model. We provide an extended discussion on other related work in Appendix A.

## 3. Preliminaries

We use the following terminology throughout this paper.

- **Procedural pretraining**: the initial "pre-pretraining" of a model to procedural data, before other stages such as **standard pretraining** with semantic data.
- **Procedural data**: non-semantic data generated from a simple algorithm, for example formal languages, cellular automata, or other rule-based processes in Section 3.2.
- **Algorithmic data**: data/tasks for diagnostic purposes to probe basic capabilities, such as needle-in-a-haystack, addition, etc. described in Section 4.
- **Semantic data**: standard (semantic) data used to pretrain language models, for example natural language, computer code, or informal mathematics.

### 3.1. Experimental Setup

We train GPT-2-type decoder-only transformers from scratch with a standard next-token prediction objective (Radford et al., 2019) (see Appendices C and E for details.). When pretraining on procedural data that involves input/output pairs (Section 3.2), we compute the loss only on output tokens. Apart from Section 6, each experiment uses a single type of procedural data.

**Data setup.** We first train each model on $T_1$ procedural tokens, then on $T_2$ standard tokens from the target data. The target data is either an algorithmic task in Section 4 for diagnostic purpose, or a semantic dataset in Section 5 reflecting standard training practices. The **baseline** is the same model trained with no procedural data ($T_1 = 0$). We adjust $T_1$ and $T_2$ following either of these two settings:

- **Additive setting.** We keep $T_2$ fixed and vary $T_1$ to measure the performance gain of *additional* procedural tokens. This evaluates whether procedural data provides a training signal that semantic data alone does not impart.
- **Substitutive setting.** We reduce $T_2$ while increasing $T_1$ (by a much smaller amount) to match the baseline performance. This evaluates how procedural pretraining can be a cheaper substitute for standard pretraining.

**Weight setup.** All the weights of the model are trained in both procedural pretraining and any subsequent training stages, i.e. nothing is frozen. Each experiment uses either of

the two following transfer settings between the two phases.

- **Full-model transfer.** The standard practice, i.e. using all procedurally-pretrained weights.[2]
- **Selective attention-only or MLP-only transfer.** We only use the pretrained weights of selected layers and reinitialize others to random values. This evaluates where useful pretrained information is stored, motivated by the evidence that MLP and attention layers perform different computations (Dong et al., 2025; Xu & Chen, 2025).

### 3.2. Generating Procedural Data

Each procedural data type is defined by a data-generating algorithm. We use algorithms that produce structurally rich data where next-token prediction requires precise symbol manipulation, compositional reasoning, and/or long-range dependency tracking. We select these from prior work and also introduce a new one (STACK). See Figure 9 for examples. They each takes hyperparameters detailed in Appendix B and generates *short sequences* (max. 128 tokens).

- **Sequence transformations.** A random sequence is presented and the model must predict its transformed version (Wu et al., 2022). This includes SET (token deduplication), REVERSE (reversing the input), IDENTITY (copying the input), UNION (ordered combination of two sequences with duplicates removed), SORT (copy in ascending order) and DELETE (removal of a specified token).
- **Memory operations.** STACK simulates a stack memory, tracking state over a random series of push and pop operations. The model must predict the final memory contents from top to bottom.
- **Formal languages.** We use classical formal languages for balanced parentheses (Hu et al., 2025; Papadimitriou & Jurafsky, 2023), K-DYCK (nested) and K-DYCK SHUF-FLE (non-nested). The model is trained for next-token prediction to generate sequences from the target language, and we vary $k$ to control the complexity of the nesting.
- **Cellular automata.** We use the elementary cellular automaton ECA RULE 110 following Zhang et al. (2024), where a binary sequence evolves via deterministic Markovian dynamics. Each sequence describes a random state of the ECA and the model must predict the next state.

## 4. Probing Procedural Pretraining with Algorithmic Reasoning

We first train small transformers (two layers, four attention heads) on specific types of procedural data, then fine-tune

---

[2]In Sections 5 and 6 we reinitialise the token embeddings to random values since there is no correspondence between the vocabularies of procedural and semantic data. In Section 4 (procedural $\rightarrow$ algorithmic transfer), we instead initialise embeddings to the mean vector, as there is no semantic domain shift.

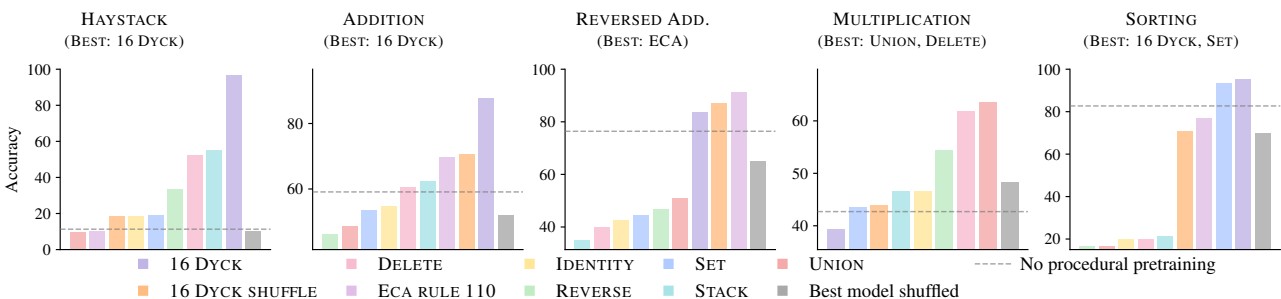

*Figure 2.* **Different types of procedural pretraining facilitate learning different algorithmic skills.** If we remove the structure within the procedural data by shuffling the sequences (*Best model shuffled*), the performance falls to the baseline. We sort the types of data by increasing performance on each task (*Best model shuffled* always last, *best performers* mentioned in the titles). Reported values are the means over 10 seeds (full results with variance in Appendix N.1).

them on common algorithmic tasks to evaluate how specific types of procedural data improve the following skills (training and test data are i.i.d.; full details are in Appendix D).

- **Memory recall.** The needle-in-a-haystack task (HAYSTACK) evaluates long-context retrieval. Each input has 30 key–value pairs ($[m_1, c_1, \ldots, m_k, c_k]$) and a query marker $m_u$; the model must output the associated value $c_u$. Accuracy is measured on the retrieved token.

- **Arithmetic.** We evaluate three tasks. ADDITION sums two 5-digit integers (a+b=), requiring right-to-left carry propagation, opposite to the autoregressive order. RE- VERSED ADDITION uses 10-digit numbers with reversed inputs and outputs, aligning carries with autoregression. MULTIPLICATION computes the product of two 5-digit integers (a×b=), predicting only result digits. All tasks are tokenized per digit, and the accuracy is measured over the output digits.

- **Logical and relational processing.** With SORTING, the model receives 10 integers from $[0, 99]$ and a separator, and outputs the sorted sequence. The accuracy is computed on the output tokens.

The model size is kept small here to avoid performance saturation through scale and to enable reliable conclusions by running each experiment with 10 different seeds.

### 4.1. Which Algorithmic Skills Improve with Procedural Pretraining?

**Setup.** We use the *additive* setting in Section 3.1: for every combination of a type of procedural data and algorithmic task, we train on $T_1$ procedural tokens then $T_2$ tokens of the algorithmic task. The baseline model uses $T_1 = 0$.

**Results.** Figure 2 shows that many types of procedural data significantly improve performance on various tasks. The best type of procedural data varies across task. For example, pretraining on K-DYCK improves context recall (HAYSTACK), while ECA RULE 110 benefits REVERSED ADDITION. This indicates that each type of procedural data

improves different skills. We also evaluate the best model pretrained on *randomly shuffled* procedural sequences. This conserves the token distribution within sequences while disrupting their structure (*Best model shuffled* in Figure 2). The performance subsequently drops back to baseline. This shows that the structure in the procedural data is essential.

> **Take-away.** Among different types of procedural data, each improves specific algorithmic skills.

### 4.2. Where does the Pretrained Information Reside?

**Setup.** We use the *selective transfer* settings defined in Section 3.1 to probe where useful information is encoded in the pretrained model. We repeat the experiments from Section 4.1 with either *attention-only* or *MLP-only transfer* and compare their performance to full-transfer to identify which component retains the most benefit.

**Results.** Figure 3 shows surprisingly that selective transfer can be superior to full-model transfer. For instance, with the IDENTITY / HAYSTACK pair, attention-only gives an 80-percentage point improvement over full-model transfer. This means that useful information is encoded in the attention layers, and that the other pretrained components (MLPs) contain non-transferable structure. Across the different tasks, the attention layers are the most consistent carrier of useful information, with the exception of REVERSED ADDITION, where MLP-only and full-model are superior.

> **Take-away.** Procedural pretraining creates localised skills in specific components of the architecture. Transferring specific components can be more effective than transferring the entire model.

### 4.3. Are There Simple Explanations for the Benefits of Pretraining?

We next examine potential mechanisms underlying the improvements from procedural pretraining. The goal is to

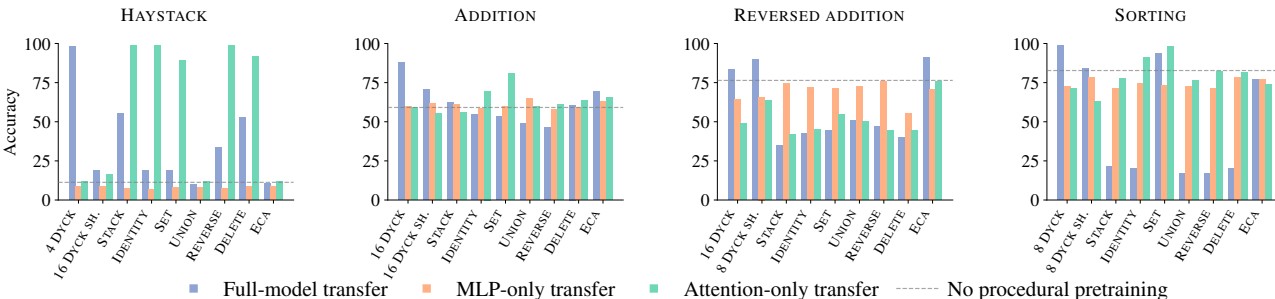

*Figure 3.* **Selective transfer of MLP or attention layers can improve over full-model transfer**, showing that procedural pretraining creates 'modular' structure localised in the selected model components. Reported values are means across 10 seeds (full results with variance in Appendix N.1).

identify whether these gains arise merely from superficial optimisation effects such as generic regularisation or weight rescaling. See Appendix F for full details and results.

**Explanation 1: attention sharpening**. We observe that pretrained models have sharp attention patterns, and transferring only the sharpest attention heads preserves or even exceeds the performance of transferring all of them. One possibility is thus that pretraining creates a generic "sharpening" of the attention (Liu et al., 2023) with no relevance to precise patterns. However, training models with an explicit regularizer for sharper attentions does not replicate the benefits of procedural pretraining. This shows that precise attention patterns do matter.

**Explanation 2: initialisation scale.** Another explanation is that pretraining simply adjusts the magnitude of initial weights (Huang et al., 2020; Wu et al., 2022). We test this using the best models from Section 4.1, and shuffle the weights per layer, such that the distributions of magnitudes are preserved but their structures erased. As expected, Figure 13 in the appendix shows that the accuracy drops dramatically. We also observe a rapid drop in accuracy with the gradual addition of Gaussian noise to the weights. This shows that pretrained weights encode meaningful structure.

> **Take-away.** The benefits of procedural pretraining are encoded in precise weight structure. They cannot be explained by a simple rescaling of the weights or generic regularisation of the attention.

## 5. Can Procedural Data Complement or Replace Standard Data?

We now examine the *practical* benefits of procedural pretraining. In Section 5.1, we use domain-specific datasets (language and code) to evaluate if the learned *abstract* algorithmic skills (Section 4) help in *semantic* domains. In Section 5.2, we turn to larger pretraining corpora including natural language mixed with code and informal math.

### 5.1. Domain-Specific Corpora

**Setup.** We use WIKITEXT (Merity et al., 2016) and Github's JAVACORPUS (Allamanis & Sutton, 2013) as domain-specific datasets of natural language and undocumented code. We train GPT-2-small models from scratch on these datasets after initial pretraining on procedural data (full-model transfer). We repeat this with different amounts of procedural tokens $T_1$ (additive setting).

**Results.** Figure 4 shows that procedural pretraining significantly outperforms the baseline for both natural language and code. Surprisingly, the improvement is not clearly correlated with the amount of procedural pretraining tokens ($T_1$) and small amounts of pretraining proves sufficient. Data generated with UNION and SET help both domains, while SORT only helps with natural language. Additional results in Appendix G show that the sequence length and the number of pretraining steps, both controlling $T_1$, influence the effectiveness of different types of procedural data.

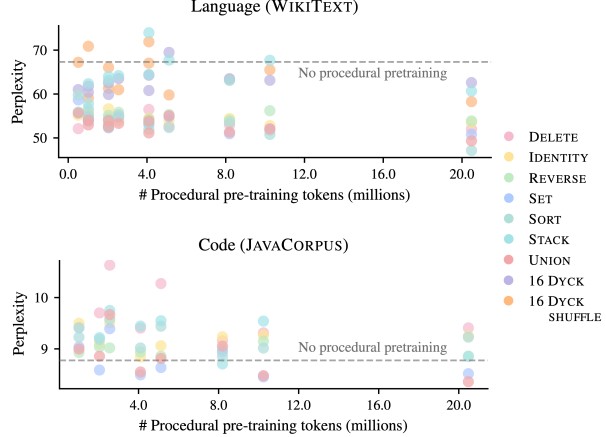

*Figure 4.* **The benefits of procedural pretraining transfer to semantic domains.** Perplexity (lower is better) on natural language (**left**) and pure code (**right**). Introducing a little procedural data is very effective: compare the number of procedural tokens ($T_1$) in these plots with the amount of tokens from the target datasets ($T_2$) being 15M for WIKITEXT and 105M for JAVACORPUS.

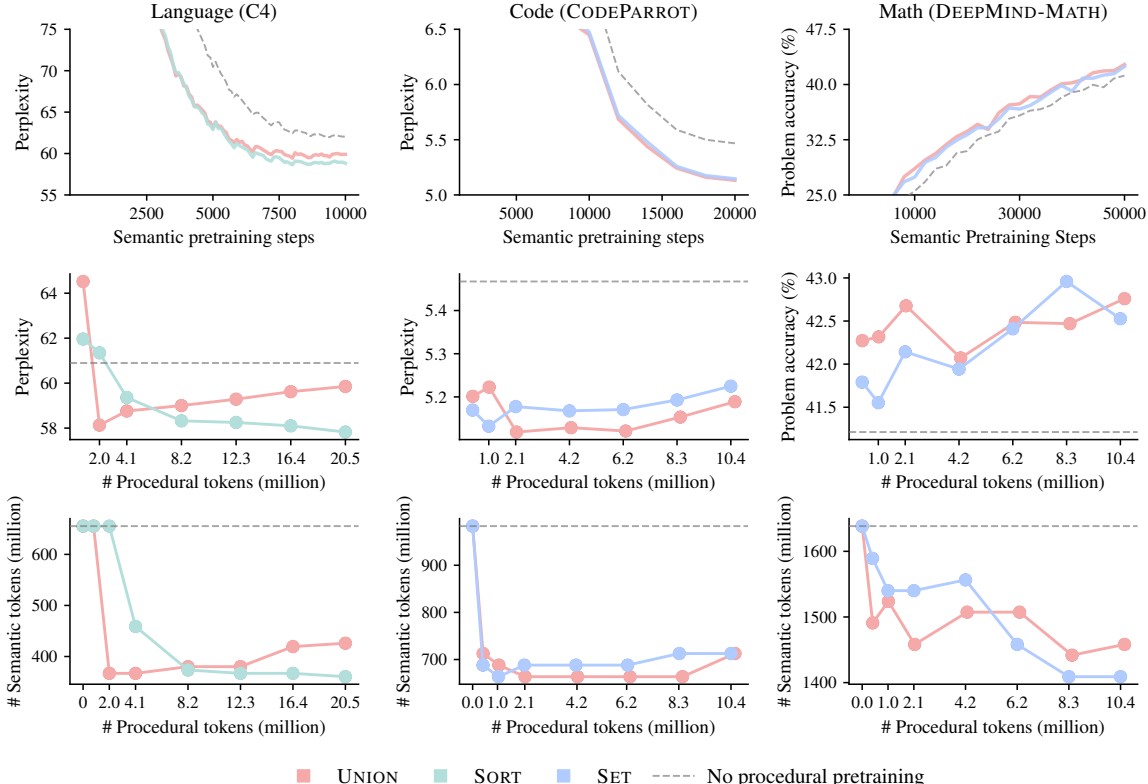

*Figure 5.* **Procedural pretraining is complementary to standard data and is highly data-efficient.** Each column corresponds to a different semantic dataset. **(Top)** Training curves with different types of procedural data (UNION, SORT, SET). **(Middle)** Additive setting: a small amount of procedural data is sufficient to outperform standard pretraining. **(Bottom)** Substitutive setting: we plot curves whose points $(x, y)$ achieve equivalent performance with $x$ procedural tokens and $y$ standard tokens. We can drastically reduce the total amount of data when using a small fraction of procedural data. Full-model transfer (see Section 3.1) is used for procedural pretraining.

---

> **Take-away.** The benefits of procedural pretraining transfer from abstract algorithmic skills to semantic domains, and they only require relatively small amounts of data.

## 5.2. Larger Pretraining Corpora

**Setup.** We expand the evaluation to more diverse and larger datasets to evaluate whether the knowledge gained from procedural pretraining is complementary to the information typically acquired from these. We use several standard pretraining datasets for **natural language** (C4, Raffel et al. (2020)), **code** (CODEPARROT, HuggingFace (2022)), and **informal mathematics** (DEEPMIND-MATH, Saxton et al. (2019), the math portion of THE PILE, Gao et al. (2020)). Much of the prior work (see Section 2) has been limited to natural language, we additionally consider informal mathematics and code because they also constitute an important part of standard pretraining corpora. We also hypothesize that they are well-suited for substantial gains from procedural pretraining due to their strong structural regularities similar to procedural data.

We use the best-performing procedural data types (UNION,

SORT, SET) identified in Figure 4 (Section 5.1). We train CodeParrot-small–style models (HuggingFace, 2022) from scratch. Each model is first pretrained on $T_1$ procedural tokens (0–20M), followed by standard pretraining on $T_2$ tokens from one of the above datasets (655M, 1B, or 1.6B). We evaluate both *additive* and *substitutive* settings. In the additive setting, we measure absolute performance gains from the $T_1$ procedural tokens. In the substitutive setting, we quantify semantic-token savings $\Delta T_2$ such that training on $(T_2 - \Delta T_2)$ semantic tokens with $T_1$ procedural tokens matches the performance of a $T_2$-only model.

**Results.** Figure 5 (top) shows that procedural pretraining accelerates and improves subsequent pretraining. The additive setting (middle) demonstrates that the benefits from procedural pretraining only require a small amount of data, and that additional data is not always beneficial. In all cases, a small amount of additional procedural tokens (2–4M) clearly outperform the baseline. For reference, 2.1M procedural tokens correspond respectively to 0.3%, 0.2%, and 0.1% of each of the three semantic datasets. The substitutive setting (bottom) shows that procedural tokens can efficiently substitute for large amounts of semantic tokens.

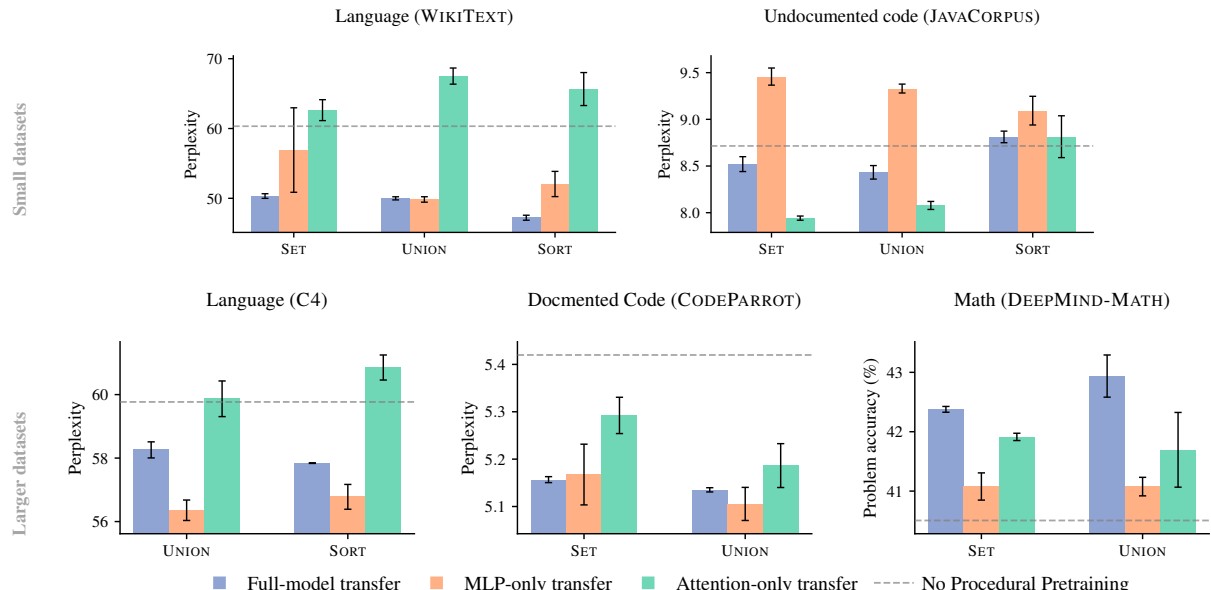

Figure 6. **Localisation of transferable pretrained information for different semantic domains**. **(Top)** Using selective weight transfer described in Section 3.1, we find that MLPs and attention layers are important respectively for natural language and pure code, across different types of procedural data. **(Bottom)** On larger datasets, MLP-only transfer works best for language. As expected, full transfer is optimal for domains involving both language and structured data (documented code, informal mathematics).

For example, with C4, we can maintain the baseline loss and save about $45\%$ of semantic tokens (and thus FLOPs) by using only $2.1$M procedural tokens. For CODEPARROT and DEEPMIND-MATH, a $33\%$ and $14\%$ tokens/FLOPs saving can be achieved with $2.1$M and $10.4$M procedural tokens.

In Appendix L, we further examine how the effects of procedural pretraining scale with both model size (350M and 1.3B parameter models) and data size (up to 10B tokens). The larger models continue to exhibit clear and consistent improvements from procedural pretraining on a larger scale.

> **Take-away.** Procedural pretraining is complementary to standard pretraining on semantic datasets in multiple domains. It is also highly data-efficient and allows one to drastically reduce the total amount of data needed to reach a given perplexity level.

### 5.3. Do the benefits persist on downstream tasks?

We further evaluate if the above benefits of procedural pretraining on the perplexity of standard pretraining transfer to downstream tasks, the primary indicator of practical model utility. Following semantic pretraining, we evaluate (in fine-tuning or few-shot regimes) both the baseline and our models on representative language (WikiText-103 (Merity et al., 2016), GLUE (Wang et al., 2019)), code completion/generation (PY150 (Lu et al., 2021), MBPP (Austin et al., 2021)) and commonsense reasoning (ARC-Easy (Clark et al., 2018), HellaSwag (Zellers et al., 2019)) datasets. As detailed in Appendix M, the improve-

ments from procedural pretraining consistently persist after downstream fine-tuning.

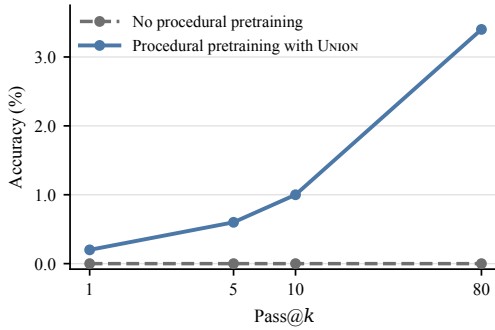

Figure 7. **The benefits of procedural pretraining transfer to downstream tasks.** Here we show 3-shot pass@$k$ problem accuracy on MBPP. See Appendix M for results on other zero-shot and fine-tuning tasks.

Notably, as shown in Figure 7, a 124M CodeParrot model with procedural pretraining starts to show non-random accuracy on MBPP, while the baseline model shows zero accuracy up to 80 attempts. Again, the compared models are identical in all respects except for the procedural warm-up, which only requires a trivial amount of extra compute.

These results further support that some of the structure that is imparted from the procedural pretraining is retained throughout subsequent semantic pretraining, echoing the finding of Jesus et al. (2021).

> **Take-away.** The benefits of procedural pretraining to standard pretraining persist after downstream fine-tuning.

### 5.4. Localisation of the Transferable Pretrained Information

**Setup.** Similar to section 4.2, we use selective weight transfer (attention-/MLP-only) to locate where the useful, transferable information resides in the procedurally-pretrained model, for the semantic domains considered so far. Note that we consider JAVACORPUS and CODEPARROT as different domains since they respectively contain pure and documented code (i.e. interleaved with natural language).

**Results.** Figure 6 shows that on JAVACORPUS (pure code), transferring only the attention layers yields the largest gains in both perplexity and code-completion accuracy (Figure 21). On WIKITEXT and C4 (natural language), the opposite holds, and transferring the MLPs is more effective than transferring attention layers. This suggests that procedural pretraining induces distinct inductive biases in different components, and selectively transferring the right component can further improve upon the results from Figure 5. For domains that combine natural language with structured data, i.e. documented code and informal math (CODEPARROT and DEEPMIND-MATH), full-model transfer performs better overall, by combining the benefits from both MLPs for natural language, and attention for structured data. The fact that transferring MLP is better than attention for natural language is intriguing given that MLPs are believed to store factual information in LLMs (Dong et al., 2025; Geva et al., 2020; Xu & Chen, 2025), raising the question of how procedural pretraining improves MLPs for handling natural language with only abstract data.

In Appendix N.2, we further explore the benefits of MLP-only transfer for language on syntactic and morphological competence. We show that MLP-only transfer achieves a better downstream accuracy on BLiMP (Warstadt et al., 2020) in the additive setting. In the substitutive setting, it requires even fewer C4 tokens to reach the same perplexity level than full-model transfer (42% vs. 55%).

> **Take-away.** Procedural pretraining instils useful transferable information in both MLPs and attention layers. MLPs benefit natural language while attention layers support structured domains such as code and mathematics.

## 6. Combining Multiple Types of Procedural Data

Our experiments and most prior work on procedural data so far use a single type of such data at a time. Combining the strengths of multiple procedural data sources is promising but not trivial due to their varying levels of learning difficulty. This section explores two techniques to combine the complementary benefits of multiple types of procedural data by building on the findings from Section 4–5.

### 6.1. Data Mixtures

**Setup.** A natural approach is to pretrain on mixtures of procedural data in chosen ratios. We evaluate pairs of procedural data $A$ and $B$ that we mix using $T_A$ and $T_B$ tokens of each, such that $T_1 = T_A + T_B$ is fixed. We prefix each pretraining sequence with an extra token specifying which of $A$ or $B$ it belongs. We train a model on these $T_1$ tokens then on $T_2$ tokens from either JAVACORPUS or WIKITEXT.

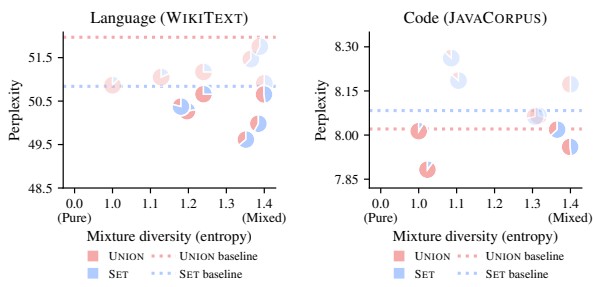

*Figure 8.* **Mixtures of two types of procedural data.** We vary the proportion of SET and UNION (indicated by the small pie charts) while keeping the total number of procedural tokens $T_1$ fixed. Some choices achieve clearly better perplexity (lower is better) than either type alone.

**Results.** Figure 8 shows that many mixtures, each with different mixture ratios shown by the pie chart and entropy of ratios, outperform the best single-source baselines for attention transfer on JAVACORPUS and full-model transfer on WIKITEXT (the best settings identified in Section 5.4). This proof of concept shows that the benefits of multiple types of data are cumulative, and suggest potential for further gains with optimized combinations of additional sources.

### 6.2. Weight Mixtures

We evaluate an alternative method that builds on the findings from Sections 4.3 and 5.4 about the localisation of pretrained information in distinct layers (attention vs. MLPs). We propose to compose a new model by assembling components from several pretrained models. This avoids the challenge of balancing data mixtures.

**Setup.** We assemble a model with the attention layers of a pretrained SET model and the MLPs of an ECA RULE 110 model. We chose these because they showed distinct and complementary capabilities (see HAYSTACK and REVERSED ADDITION in Table 1). We then further train this model on the algorithmic evaluation tasks of Section 4.

**Results.** The last row of Table 1 shows that the combined model yields superior performance across the four tasks,

while the single-source models have weaknesses on one or more tasks. This indicates that procedurally-pretrained models can be modularly combined by simply assembling their most useful components.

*Table 1.* **Procedurally-pretrained models combined at the weight level.** We combine SET-pretrained attention layers with ECA-pretrained MLPs (last row). This yields strong performance across all four tasks, whereas single-source models show weaknesses in at least one task. Full results with variance in Table 14.

|  | HAYSTACK | ADDITION | REVERSED ADDITION | SORT | AVG. |
|---|---|---|---|---|---|
| No procedural pretraining | 11.3 | 59.1 | 76.4 | 82.7 | 57.4 |
| SET (full-model transfer) | 18.9 | 53.4 | 44.6 | 93.5 | 52.6 |
| SET (attention-only transfer) | 88.9 | **81.1** | 54.4 | 98.1 | 80.6 |
| ECA (full-model transfer) | 10.5 | 69.6 | **91.0** | 76.9 | 62.0 |
| ECA (MLP-only transfer) | 8.71 | 63.1 | 70.5 | 77.1 | 54.9 |
| SET (attention) + ECA (MLP) | **94.4** | 80.3 | 82.9 | **99.4** | **89.3** |

> **Take-away.** The effects of multiple types of procedural data are additive. Proof-of-concept experiments show that they can be combined both at data and weight levels, and suggest ample room for further benefits with larger and more-optimised combinations.

## 7. Discussion

This paper shows that pretraining language models on well-chosen abstract procedural data complements standard pretraining, accelerating training and improving performance on natural language, code, and informal mathematics. Our experiments also shed light on the origin of these gains. We found that useful information lies in different components (MLP vs. attention) depending on the domain (language vs. structured domains). These findings motivate new pretraining paradigms where primitive abstract data is exposed to LLMs before they acquire rich world knowledge.

**Efficient initialisation.** Unlike standard data, procedural data has a small Kolmogorov complexity, meaning that it contains information that can be summarized in a few lines of code. In principle, it may be possible to simplify this as a deterministic or closed-form *smart initialisation* of LLMs.

**Why is procedural data helpful?** Our results in Section 4.3 rule out simple explanations, indicating deeper effects than merely a better optimisation dynamics or memorisation. Investigating a first-principles explanation or studying the mechanisms at play using mechanistic interpretability techniques (Conmy et al., 2023) is a promising avenue.

**Combining multiple types of procedural data.** We showed that the benefits can be additive. Existing methods for data mixture optimization (Fan et al., 2023; Xie et al., 2023; 2025) could be adapted to optimally balance multiple types of procedural data.

**Knowledge vs. reasoning.** Han et al. (2025) argue that LLMs' limitations stem from entangled representations of knowledge and reasoning. Our work can be viewed as injecting an 'algorithmic reasoning prior' before world-knowledge acquisition. This ultimately suggests a data-driven path in improving knowledge and reasoning acquisition beyond architectural changes (Pouransari et al., 2025).

**Limitations.** (1) We use smaller models (up to 1.3B) and lower data-model ratio compared to state-of-the-art LLMs, further scaling up our experiments is an important future step. (2) While our experiments on combining multiple types of procedural data are a proof of concept, they lay out several promising directions for future work.

## Impact Statement

This paper presents work whose goal is to advance the field of Machine Learning. There are many potential societal consequences of our work, none of which we feel must be specifically highlighted here.

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

# Appendix

The appendix provides the following additional details and results:

## A. Extended Literature Review

**What is learned by pretraining language models.** The quantity (Kaplan et al., 2020) and quality (Longpre et al., 2024) of pretraining data are empirically critical for the performance of large language models. But recent results also question the value of the data, showing that some benefits of pretraining are attributable to the optimisation objective more than the actual data. Balestriero & Huang (2024) compared models trained for text classification from random initialisation with fine-tuning from a pretrained checkpoint. They found that pretraining provides little benefit for tasks that do not involve text generation. Krishna et al. (2023) showed success in re-using the same data for pretraining and fine-tuning, showing also that the pretraining objective matters more than the data being used. The same conclusion follows from results of pretraining on synthetic data devoid of semantic meaning, e.g. for machine translation (He et al., 2023), computer vision (Baradad et al., 2021), visual navigation (Wang et al., 2022), and reinforcement learning (Baradad et al., 2022). This paper examines such purely synthetic pretraining to understand the exact capabilities that can be obtained from procedurally-generated data.

**What matters in pretraining data.** The selection of data to pretrain frontier models mostly relies on experimentation (Longpre et al., 2024). However, several key distributional and structural properties of the data have also been identified, such as data repetition to foster generalisation (Charton & Kempe, 2024) and burstiness to enable in-context learning (Chan et al., 2022). Computer code is empirically very effective as pretraining data for LLMs, as it improves their abilities for compositional generalisation and math-related tasks (Aryabumi et al., 2024; Petty et al., 2024). This presumably results from the abundant compositional and recursive patterns in computer code, but a better understanding of the mechanisms at play is lacking to reap the full benefits of structure in pretraining data. In this paper, we replicate the positive effects of structured pretraining data in controlled settings, and study how such data imparts useful inductive biases to the model.

**Pretraining on procedural data.** Most attempts to train language models with synthetic data follow a linguistic perspective, using formal languages to imitate properties of natural language (Chiang & Lee, 2022; Goodale et al., 2025; McCoy & Griffiths, 2023; Papadimitriou & Jurafsky, 2023; Ri & Tsuruoka, 2022). Recent work considers increasingly simpler forms of synthetic data such as input/outputs of simple algorithms (Lindemann et al., 2024; Wu et al., 2022). In these papers, specific forms of synthetic pretraining data prove helpful to subsequent fine-tuning on natural language tasks. Hu et al. (2025) provide strong empirical benefits, showing that data generated from formal languages is more valuable token-per-token than natural language for training a 1B-parameter language model. Zhang et al. (2024) pretrain on traces of cellular automata and show marginal but consistent improvements on simple reasoning tasks. Our study complements this line of work by examining more closely the pretrained models on diagnostic tasks, rather than evaluating their general handling of natural language. We identify specific capabilities imparted by specific types of procedural tasks, and locate useful structure in different parts of the architecture. We also investigate methods to combine the benefits from multiple complementary tasks.

**Procedural data in vision and RL.**    Vision transformers (ViTs) have been trained on synthetic data of increasingly simple nature (Baradad et al., 2021). Nakamura et al. (2024) pretrained ViTs on a single fractal image with augmentations that remarkably match the performance of ImageNet-pretrained models after fine-tuning. This indicates that structural properties of the data matter more than its semantic contents. Similar results exist in reinforcement learning with models pretrained on data generated from random Markov chains (Wang et al., 2023) and noise-based images (Baradad et al., 2022).

**Partial transfer from pretrained transformers.**    Zhang et al. (2023) and (Xu et al., 2023) showed that copying subsets of pretrained weights could transfer specific capabilities. Abnar et al. (2020) used knowledge distillation to transfer the inductive biases of one architecture into another. The "mimetic initialisation" of self-attention (Trockman & Kolter, 2023) is a procedure handcrafted to imitate the locality bias of pretrained models. We also evaluate the partial transfer of pretrained weights, which reveals that different pretraining tasks create useful structure in different parts of the architecture.

**Pretraining as an inductive bias.**    Pretraining transformers on synthetic data has been used to mimic the inductive biases of Bayesian inference (Müller et al., 2021) and Solomonoff Induction (Grau-Moya et al., 2024). Goodale et al. (2025) showed that well-chosen formal languages can teach complex mechanisms (e.g. counters) to a sequence model. Pretraining can generally be seen as a *soft* inductive bias for subsequent fine-tuning. But there is a large gap in our understanding of its effects compared to those of *hard* inductive biases of neural architectures (Teney et al., 2024; 2025). Han et al. (2025) argue that the difficulties of LLMs to reason robustly is due to their entangled representation of knowledge and reasoning. Much remains to be understood about how both are learned from the same data (Ruis et al., 2024). Our results suggest that procedural data could be one way to acquire reasoning mechanisms independently from specific pieces of knowledge.

## B. Procedural Pretraining

| Pretraining task | Example sequence |
|---|---|
| $k$-DYCK | ( [ { } ] ) |
| $k$-DYCK SHUFFLE | ( [ { ] ) } |
| STACK | 1 2 3 P ǀ **2 1** |
| IDENTITY | 1 2 3 ǀ **1 2 3** |
| SET | 1 2 2 ǀ **1 2** |
| SORT | 3 1 2 ǀ **1 2 3** |
| REVERSE | 1 2 3 ǀ **3 2 1** |
| UNION | 1 2 ǀ 2 3 ǀ **1 2 3** |
| DELETE | 1 2 3 ǀ 2 ǀ **1 3** |

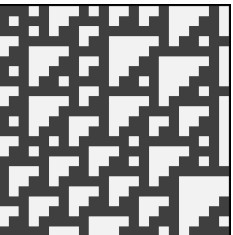

*Figure 9.* We pretrain transformers on various forms of procedural data generated from simple algorithms, such as formal languages (left) or elementary cellular automata (right). In $k$-DYCK examples, matching brackets are color-coded. For STACK, 'P' denotes the *pop* operation. The symbol 'ǀ' acts as a delimiter between the input and the expected output, on which the loss is computed (**bold** tokens). For UNION and DELETE, the first delimiter separates the two sequences to which the transformation is applied, and the second delimiter separates the entire input from the target output.

**Sequence Transformations and Memory Operations Input Sequence Lengths.**

For the sequence transformation and memory operation tasks in Section 4, procedural pretraining follows a curriculum learning scheme: models begin with input sequences of length 2 or 4 (depending on the task), and the length is increased by 2 once 99% accuracy is achieved, continuing until a maximum length of 20.

In Section 5, larger transformers are instead pretrained on procedural tasks with fixed input lengths of 8, 16, 32, and 64. Appendix G analyses the effect of sequence length, while Appendix H examines the impact of extending lengths further.

For consistency in token counts, we assume the output sequence is at most twice the length of the input, and thus estimate and report the total number of procedural tokens as $2\times$ the input length.

**Sequence Transformation Descriptions.**

IDENTITY. The input is a sequence of tokens followed by a separator. The target is an exact copy of the input sequence. The vocabulary has 102 tokens: 100 valid elements, one separator, and one padding token.

SET. The input is a sequence of tokens followed by a separator. The target is the same sequence with duplicates removed, preserving the order of first appearance. The vocabulary has 102 tokens: 100 valid elements, one separator, and one padding token.

UNION. The input consists of two token sequences separated by a delimiter. The target is the union of both sequences, preserving the order of first appearance. The vocabulary has 103 tokens: 100 valid elements, one separator, one padding token, and one union delimiter.

DELETE. The input is a sequence of tokens followed by a separator and a designated token. The target is the sequence with all instances of the designated token removed. The vocabulary has 103 tokens: 100 valid elements, one separator, one padding token, and one delete marker.

SORT. The input is a random sequence of tokens followed by a separator. The target is the same sequence sorted in ascending numerical order. The vocabulary has 102 tokens: 100 valid elements, one separator, and one padding token.

REVERSE. The input is a sequence of tokens followed by a separator. The target is the same sequence in reverse order. The vocabulary has 102 tokens: 100 valid elements, one separator, and one padding token.

**Memory Operation Descriptions.**

STACK. The input encodes a sequence of `push` and `pop` operations, followed by a separator. The target is the final stack contents, listed top-to-bottom. Tokens are pushed with 75% probability in the first two thirds of the input and popped with 75% probability in the final third. Each push inserts a unique token, pops remove the top element, and only one copy of a token may exist on the stack at any time. The vocabulary has 103 tokens: 100 pushable elements, one `pop` token, one separator, and one padding token.

**Other Procedural Data Source Descriptions.**

$k$-DYCK. We generate sequences of correctly nested parentheses using $k$ distinct bracket pairs (vocabulary size $2k$), with $k \in \{4, 8, 16\}$. All training sequences are fixed to length 128 and constructed incrementally via a stack-based procedure ensuring syntactic validity. At each step, the generator samples an opening or closing bracket with probability $p_{\text{open}} = 0.49$ (Papadimitriou & Jurafsky, 2023), forcing closure when the remaining token budget matches the number of open brackets.

$k$-DYCK SHUFFLE. This variant retains the same $2k$-token vocabulary of bracket pairs but removes the requirement of proper nesting. Sequences are sampled with a 50% probability of opening brackets and fixed to length 128, with $k \in \{4, 8, 16\}$. While every opening bracket is eventually closed, truncation can yield ill-formed strings (Hu et al., 2025), though we did not observe adverse effects in practice.

ECA RULE 110. We follow the setup of Zhang et al. (2024), generating data from Elementary Cellular Automata under Rule 110, a Class IV system with Turing-complete dynamics. To model binary state sequences with GPT-2, the embedding layer is replaced by a linear projection from binary vectors, and the output softmax is replaced by a projection back to binary space, preserving determinism. For transfer, we average the learned input embeddings across the ECA data and use this representation to initialize the embedding layers of downstream transformers.

## C. Model details

We use a GPT-2-type architecture (Radford et al., 2019) throughout our experiments. In Section 4, we employ a minimal configuration with 2 layers, 4 attention heads, and a hidden size of 16 for HAYSTACK, ADDITION, REVERSED ADDITION and SORTING. For MULTIPLICATION, we use a model size of 4 layers, 8 attention heads and a hidden size of 512. In Section 5 and 6, we use the `small` GPT-2 variant with 12 layers, 12 attention heads, and a hidden dimension of 768.

## D. Algorithmic Task Descriptions

**Memory Recall.**

HAYSTACK. This task tests a model's ability to retrieve information from long sequences. Each input consists of a sequence of key–value pairs of the form $[m_1, c_1, m_2, c_2, \ldots, m_k, c_k, m_u]$, where each $m_i$ is a unique marker and $c_i$ its associated value. The sequence terminates with a query marker $m_u$, and the model must locate its earlier occurrence in the context and output the corresponding value $c_u$. We fix $k = 30$ in all experiments and report accuracy based on whether the predicted value matches $c_u$.

**Arithmetic.**

ADDITION. This task probes a model's ability to learn the compositional structure of arithmetic addition when expressed in *forward* (non-reversed) notation. In this setting, the least significant digits, crucial for carry operations, appear at the *end* of the sequence. As a result, transformers must propagate carry information *backward* through the context, a dependency pattern misaligned with the autoregressive training objective. Each input takes the form a+b=, where $a$ and $b$ are randomly sampled $n$-digit integers. Inputs and outputs are digit-tokenized, with operator symbols (+, =) assigned unique tokens. The model is trained to predict only the result digits, and cross-entropy loss is computed solely on these positions. For all experiments we fix $n = 5$, and report token-level accuracy on the predicted sum.

REVERSED ADDITION. This variant evaluates the same underlying arithmetic skill as ADDITION, but aligns the sequence structure with the autoregression of the transformer. Both input and output sequences are reversed, so carry propagation proceeds left-to-right in the same direction as generation. For example, the sum $ab + cd = efg$ is represented as input b a d c with output g f e. The task reduces long-range dependencies while preserving the need for multi-step reasoning. We set $n = 10$ and evaluate using token-level accuracy.

MULTIPLICATION. This task evaluates a model's ability to perform multi-digit multiplication. Each input takes the form $a \times b =$, where $a$ and $b$ are randomly sampled $n$-digit integers. The model must generate the digit sequence corresponding to their product. Inputs and outputs are tokenized at the digit level, with the multiplication operator ($\times$) and equals sign (=) assigned special tokens. For all experiments we fix $n = 5$. Cross-entropy loss and token-level accuracy are computed only on the output positions corresponding to the product digits.

**Logical and relational processing.**

SORTING. This task assesses a model's ability to perform algorithmic reasoning by sorting a sequence of integers. Each input consists of a list of $n$ integers sampled uniformly from the range $[0, P - 1]$, where $P$ denotes the vocabulary size. We fix $n = 10$ and $P = 100$. The input sequence is followed by a separator token, after which the model must output the sorted version of the sequence. For example, the input 6 3 5 | requires the output 3 5 6. Training is autoregressive, and evaluation is performed only on the output positions following the separator, with token-level accuracy as the metric.

# E. Experimental Details

## E.1. Procedural Pretraining

### Details for Section 4.

The hyperparameters used for procedural pretraining are summarised in Table 2, with the exception of ECA RULE 110, whose configuration is reported separately below.

| Task | SEQ. LENGTH | LEARNING RATE | VOCAB. SIZE |
|---|---|---|---|
| IDENTITY | 4–20 | $5 \times 10^{-4}$ | 102 |
| SET | 2–20 | $5 \times 10^{-4}$ | 102 |
| STACK | 4–20 | $5 \times 10^{-4}$ | 103 |
| $k$-DYCK | 128 | $5 \times 10^{-5}$ | $2 \times k$ |
| $k$-DYCK SHUFFLE | 128 | $5 \times 10^{-5}$ | $2 \times k$ |

*Table 2.* Pretraining hyperparameters for each procedural task. All models use AdamW with weight decay 0.01, batch size 256, and run for 1,000,000 steps. Early stopping (100 validation checks) is applied for the algorithmic tasks.

ECA RULE 110. Following Zhang et al. (2024), we pretrain models on data procedurally generated from Elementary Cellular Automata under Rule 110. Each epoch begins from a new random initial state, ensuring continual access to fresh samples and effectively unlimited training data. Models are trained for up to 10,000 epochs with early stopping on validation loss. We use Adam with a learning rate of $2 \times 10^{-6}$, weight decay 0.01, and gradient clipping at norm 1.0, with batch size 64 (60 time steps, 100 spatial dimensions). The learning rate schedule consists of a 10% warm-up phase followed by cosine decay.

### Detail for Section 5.

For all algorithmic procedural tasks used in this section (IDENTITY, SET, UNION, DELETE, SORT, REVERSE, and STACK), we train using AdamW with a batch size of 64 and no warmup steps. Following Hu et al. (2025), we pretrain models

on procedural data with a weight decay of 0.1 for WIKITEXT and C4, and use 0.01 for JAVACORPUS, CODEPARROT, and DEEPMIND-MATH. The pretrained models are subsequently fine-tuned on their respective downstream datasets. An ablation study in Appendix K confirms that this choice of weight decay during pretraining does not affect our conclusions. We sweep sequence lengths over $\{8, 16, 32, 64\}$ and vary the number of procedural pretraining steps between 100 and 2500. No warmup or learning rate decay is applied; instead, we train with a fixed learning rate throughout. For consistency, the learning rate during pretraining is matched to that of the downstream semantic objective, as preliminary experiments indicated this setting to be most effective.

### E.2. Algorithmic Tasks

HAYSTACK, FORWARD ADDITION, REVERSED ADDITION, and SORTING. We trained models for $10^4$ steps with a batch size of 1,000. The training data is generated dynamically. We used the AdamW optimizer with a learning rate of $10^{-3}$ and weight decay of $10^{-3}$. We always use an architecture consisting of 2 layers, 4 attention heads, and 16-dimensional embeddings. We report mean and standard deviation over 10 seeds in Appendix N.

MULTIPLICATION. These experiments employed a larger model with 4 layers, 8 attention heads, and 512-dimensional embeddings. Thus, we use a smaller training batch size (64 vs. 1,000), resulting in approximately 156k update steps compared to 10k steps for the afforementioned reasoning tasks, despite using the same number of training examples. We optimize with AdamW using a learning rate of $10^{-3}$, weight decay of $10^{-3}$, and 500 warmup steps. We run this over 3 seeds, and report standard deviations in Appendix N.

### E.3. Semantic Data

WIKITEXT. We train our models on Wikitext-2 (Merity et al., 2016) using next-token prediction with AdamW. Training runs for $\sim$7 epochs (5,000 steps) with an effective batch size of 32. We use a learning rate of $5 \times 10^{-4}$ with cosine decay and no warmup steps. Sequences are tokenized with the GPT-2 tokenizer, truncated to 1,024 tokens. We evaluate the model on the validation split, using 1,024 samples. Our primary metric is validation perplexity.

JAVACORPUS: We train our models on Github's JavaCorpus (Allamanis & Sutton, 2013) using next-token prediction with AdamW. Training runs for 5 epochs with an effective batch size of 8. We use a learning rate of $8 \times 10^{-5}$ and no warmup steps. The hyperparameters follow those in (Lu et al., 2021). Sequences are tokenized with the CodeGPT (Lu et al., 2021) tokenizer, with block size 1,024 tokens. We report validation perplexity and test accuracy for code completion.

C4: We pretrain our models on the C4 dataset (Raffel et al., 2020) using next-token prediction with AdamW. Training runs for 10,000 steps with an effective batch size of 32. We use a learning rate of $5 \times 10^{-4}$ with cosine decay and no warmup steps. Sequences are tokenized with the GPT-2 tokenizer and truncated to 2,048 tokens. We evaluate models on the C4 validation split using 1,024 samples, reporting validation perplexity. To assess linguistic generalization, we also report accuracy on the BLiMP grammaticality judgment benchmark (Warstadt et al., 2020), which tests whether models prefer grammatical over ungrammatical sentence pairs.

CODEPARROT: We pretrain our models on the CodeParrot dataset[3] using next-token prediction with AdamW. Training runs for 20,000 steps with an effective batch size of 48. We use a learning rate of $5 \times 10^{-4}$ with cosine decay, no warmup steps, and weight decay of 0.1. Sequences are tokenized with the CodeParrot's tokenizer and with length 1,024 tokens. We evaluate models on the validation split with 1,000 evaluation steps and a batch size of 48, reporting validation loss and perplexity.

DEEPMIND-MATH: We pretrain our models on the Deepmind-Mathematics dataset (Saxton et al., 2019) using next-token prediction with AdamW. Training runs for 50,000 steps with an effective batch size of 64. We use a constant learning rate of $8 \times 10^{-5}$ (as is done in the original paper), no warmup steps, and weight decay of 0.1. Sequences are tokenized at the character-level (including digits, alphabet in upper and lower case, punctuation and whitespace, a total of 95 different tokens) and have a length 512 tokens. We evaluate models on the in-distribution validation split with 100 evaluation steps and a batch size of 64, reporting the accuracy on the validation problems. This ensures evaluating around 38,000 questions in each validation session. A problem is considered correct if and only if the prediction exactly matches the groundtruth answer.

For the 124M-parameter models trained on C4 and CODEPARROT, we follow the training recipe of HuggingFace's CodeParrot-small model and the setup used by Hu et al. (2025). For experiments on DEEPMIND-MATH, we additionally

---

[3] https://huggingface.co/datasets/codeparrot/codeparrot-clean

follow the settings from the original dataset paper (Saxton et al., 2019). For the larger 350M- and 1.3B-parameter models, we use the architectures and learning rates from the EleutherAI Pythia suite (Biderman et al., 2023).

### E.4. Downstream Finetuning

WIKITEXT-103: We finetune our language models on the WIKITEXT-103 dataset (Merity et al., 2016). Finetuning runs for ~37 million tokens (10,000 steps) with an effective batch size of 32. We use a learning rate of $1 \times 10^{-4}$ with cosine decay and no warmup steps. Sequences are tokenized with the GPT-2 tokenizer, truncated to 2,048 tokens.

GLUE: We finetune our language models on the GLUE benchmark (Wang et al., 2019). For all evalautions, fine-tuning is run for one epoch with a batch size of 16 and learning rate of $5 \times 10^{-5}$ with a linear decay.

PY150: We finetune our models on PY150 (Raychev et al., 2016), which is an influential task evaluating code completion capability (Lu et al., 2021). It contains 150,000 Python source files collected from GitHub. We first follow Lu et al. (2021) for the preprocessing and then finetune the models using next-token prediction with AdamW. Training runs for 2 epochs with an effective batch size of 8, a learning rate of $8 \times 10^{-5}$, and a 0.01 weight decay. Sequences are tokenized with the CodeGPT (Lu et al., 2021) tokenizer, with block size 1,024 tokens. We report test accuracy (token-level accuracy) on this task.

## F. Testing Simpler Explanations

### F.1. Attention Sharpening

This appendix analyses whether the benefits of procedural pretraining arise from generic attention sharpening. First, we find that a small subset of sharpened attention heads contain the useful inductive bias for downstream tasks. Then, we attempt to reproduce the behaviour of these heads through regularisation. We find this does not provide the same benefits, demonstrating that procedural pretraining fosters specific inductive biases beyond generic attention sharpening.

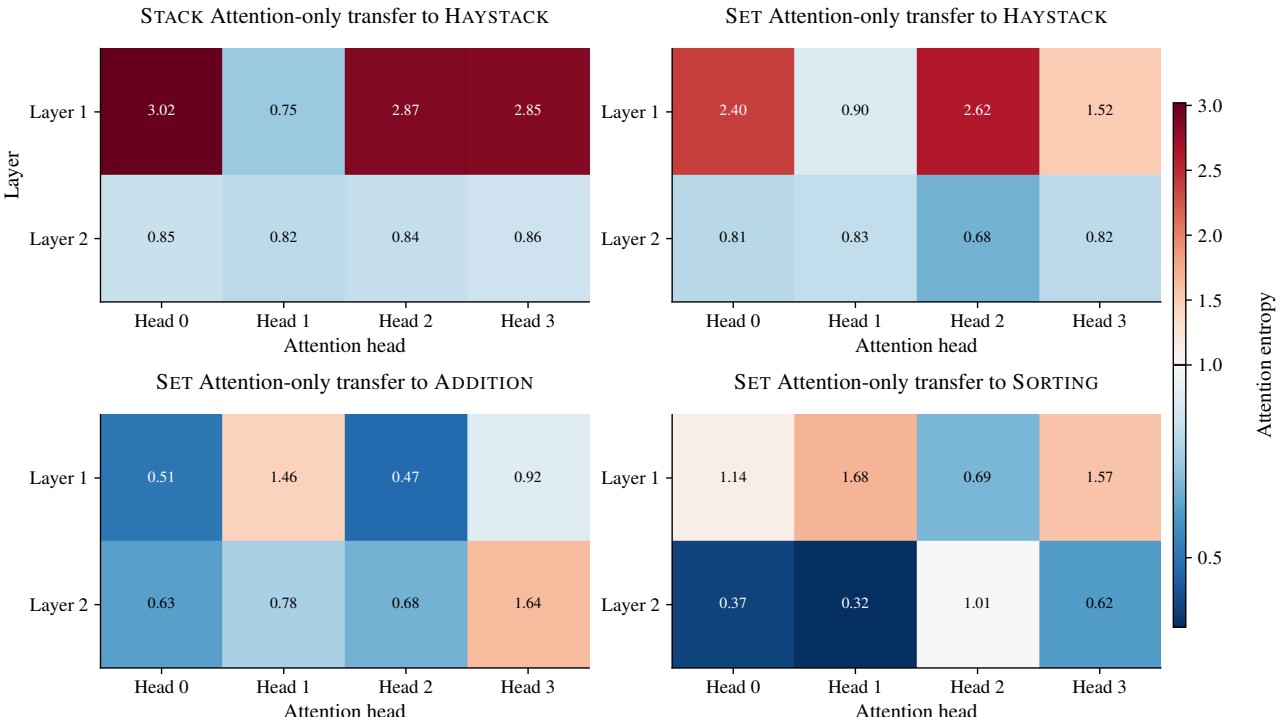

*Figure 10.* Head-wise attention entropy after fine-tuning. Procedural pretraining yields a subset of low-entropy heads (blue).

### F.1.1. ATTENTION ENTROPY ANALYSIS

We first examine the attention patterns of the procedurally pretrained models after fine-tuning on the downstream tasks.

**Setup.** We measure the sharpness of each attention head using entropy,

$$H = -\sum_i p_i \log p_i,$$

where $p_i$ denotes the normalized attention weight assigned to token $i$. Low entropy corresponds to selective attention, while high entropy reflects diffuse, uniform distributions. We compute head-wise entropy after fine-tuning, averaging over 100 downstream evaluation examples.

**Results.** Figure 10 shows that procedural pretraining leads models, after downstream fine-tuning, to consistently develop a subset of low-entropy heads. For example, a STACK-pretrained model fine-tuned on HAYSTACK exhibits five of eight heads with entropy close to $H \approx 0.8$, while the remaining three have substantially higher entropy around $H \approx 3.0$.

### F.1.2. SELECTIVE TRANSFER OF LOW-ENTROPY HEADS

We hypothesise that the useful inductive biases introduced by procedural pretraining are concentrated in the subset of low-entropy attention heads.

**Setup.** To test our hypothesis, we fine-tune on the downstream task while transferring either the three lowest-entropy heads that emerge from the procedurally pretrained model (identified post hoc after finetuning) or, for comparison, the three highest-entropy heads.

**Results.** Figure 11 shows that transferring only the three lowest-entropy heads preserves, and in some cases even surpases the performance of full attention transfer. In contrast, transferring the three highest-entropy heads results in performance comparable to the baseline without procedurally pretrained attention. These results demonstrate that the benefits of procedural pretraining can be concentrated in a small subset of sharp, low-entropy attention heads.

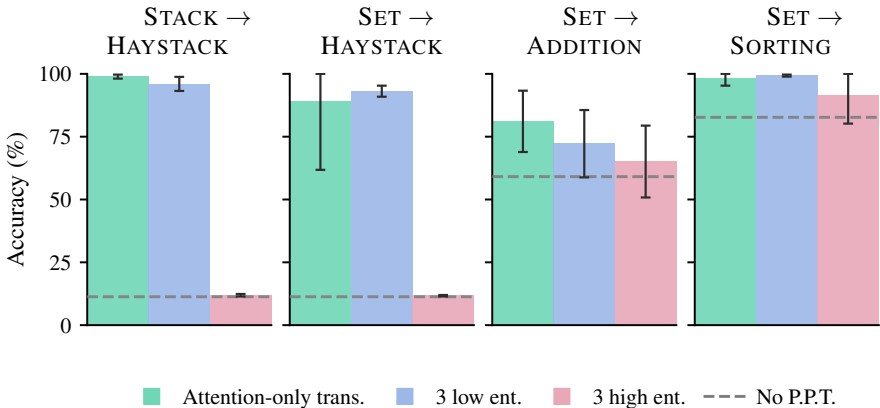

*Figure 11.* Validation accuracy after downstream fine-tuning when transferring subsets of procedurally pretrained attention heads. The three lowest-entropy heads preserve or even surpass full transfer, while the three highest-entropy heads perform comparably to a baseline without procedural pretraining. Results are over 10 random seeds.

### F.1.3. ENTROPY REGULARISATION TO SELECTED ATTENTION HEADS

We next investigate whether the benefits of procedural pretraining can be reproduced by explicitly enforcing low-entropy attention.

**Setup.** We attempt to replicate the behavior of the beneficial attention heads through regularization. An entropy regularization term is introduced during finetuning on HAYSTACK to a model that did not undergo procedural pretraining. This regularization is applied to three selected heads and drives them toward a target entropy of $\tau = 0.8$, matching the

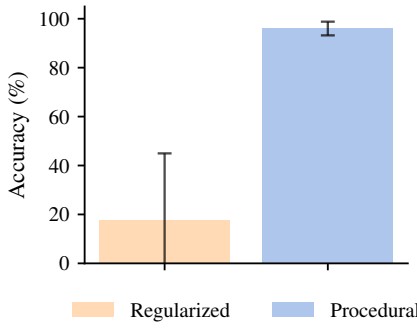

*Figure 12.* **Validation accuracy on HAYSTACK with entropy regularisation.** Models trained from scratch with explicitly enforced low-entropy heads (orange) underperform those with procedurally pretrained heads (blue), indicating that sharper attention alone is insufficient. Results are averaged over 10 random seeds.

average entropy observed in the three heads shown to carry useful inductive biases from STACK pretraining (Figure 10 and Figure 11).

**Results.** As shown in Figure 12, this approach is ineffective: the regularized heads perform substantially worse than the STACK-pretrained heads when evaluated on the HAYSTACK task.

In summary, these findings indicate that simply enforcing sharper attention is insufficient to reproduce the benefits of procedural pretraining. Low entropy appears to be a side effect of the inductive biases acquired through procedural pretraining rather than the cause of improved performance.

### F.2. Weight Scaling

We test whether the benefits of procedural pretraining arise solely from weight distribution adjustments, as opposed to precise weight structures and values. Our results show that the gains depend critically on the latter.

**Weight Shuffling.** We apply layer-wise shuffling of the pretrained weights to the best-performing models from Section 4.2 and evaluate downstream accuracy after fine-tuning. This setup explicitly preserves weight distributions while erasing structure. Figure 13 demonstrates that weight distributions alone are insufficient: performance collapses to the no procedural pretraining baseline, except for SORTING, which retains partial benefits. We use 10 seeds and report mean results, with variance data in Appendix N.

**Noise Injection.** We introduce additive Gaussian noise to the procedurally pretrained weights of the best models from Section 4.2 and evaluate performance after fine-tuning. We report a relative improvement score, where 1.0 corresponds to unperturbed pretrained weights and 0.0 corresponds to a baseline without procedural pretraining (random initialisation). Figure 14 shows that gradually increasing Gaussian noise consistently degrades performance, confirming that precise weight values are crucial. We use 10 seeds and report mean results, with variance data in Appendix N.

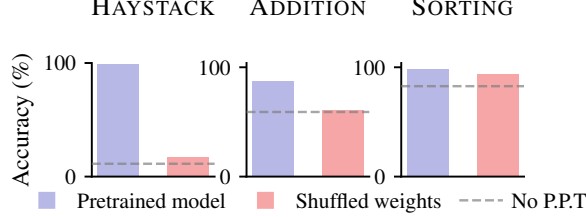

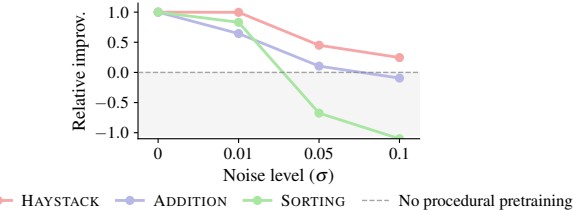

*Figure 13.* Layer-wise weight shuffling largely eliminates the benefits of procedural pretraining, despite preserving the overall distribution of weight values. This indicates that the advantages arise from precise structural organisation of the weights, rather than from their distribution alone.

*Figure 14.* Injecting Gaussian noise into pretrained weights progressively erodes the benefits of procedural pretraining. This demonstrates that precise weight values are essential, and coarse statistics such as weight magnitudes alone cannot account for the performance benefits.

## G. Procedural Data Hyperparameter Grid Search

We study the influence of both pretraining steps and input sequence length on the effectiveness of procedural pretraining for downstream semantic tasks.

**Setup.** We conduct a grid search over sequence length and number of pretraining steps to determine which configurations of procedural pretraining yield the lowest semantic validation perplexity. Each model is first pretrained on a single procedural task for $T_1$ tokens, followed by $T_2$ tokens of semantic data (WIKITEXT for natural language and JAVACORPUS for code), with full-model transfer. The value of $T_1$ is varied by adjusting the sequence length and number of pretraining steps, while $T_2$ remains fixed.

**Results.** Figure 15 and 16 report validation perplexity across all configurations, showing that both sequence length and pretraining steps strongly influence performance, with optimal settings differing by domain and task.

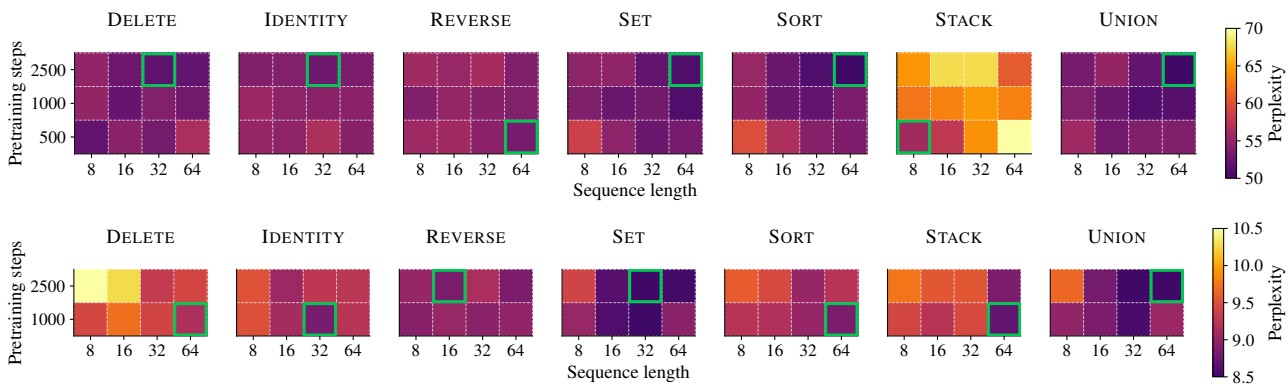

*Figure 15.* Validation perplexity for different configurations of procedural pretraining when finetuned on WIKITEXT (top) and JAVACORPUS (bottom), sweeping over sequence length and number of pretraining steps. Each panel corresponds to a distinct procedural task, with colours indicating perplexity (lower is better). The best-performing configuration for each task is marked in green.

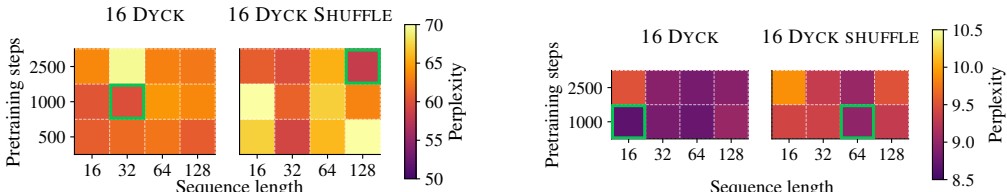

*Figure 16.* Validation perplexity for DYCK and DYCK SHUFFLE procedural pretraining when fine-tuned on WIKITEXT (left) and JAVACORPUS (right), sweeping over sequence length and number of pretraining steps. Setup matches Figure 15. Colours indicate perplexity (lower is better), with the best-performing configuration marked in green.

## H. Longer Sequences for Procedural Pretraining

We extend the sequence length search on WIKITEXT from 8–64 tokens (Appendix G) to 128 tokens using full-model transfer for the best perfoming procedural tasks. Results are mixed: SET benefits from longer sequences, while SORT and UNION do not. Thus, the utility of longer procedural sequences is task-dependent.

## I. Transferability Analysis

We analyse the correlation between procedural pretraining loss and downstream loss on C4. For SET and UNION, transfer performance deteriorates when procedural loss is either too high or too low, suggesting that both underfitting and overfitting impair generalization. Consequently, the strongest transfer is observed at intermediate levels of procedural optimization. In contrast, for SORT, transfer performance contintues to improve steadily as procedural loss decreases, demonstrating that the transferability of procedural pretraining is task dependent.

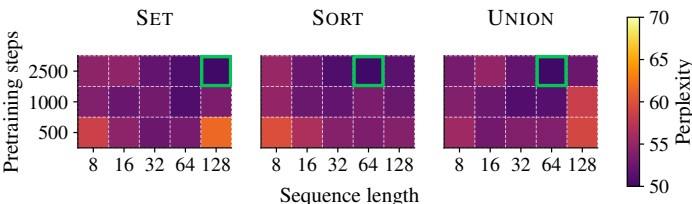

*Figure 17.* Effect of extending sequence length during procedural pretraining on WIKITEXT. Longer sequences improve subsequent language modelling for SET but not SORT or UNION, showing that the benefit of extended contexts is task-dependent.

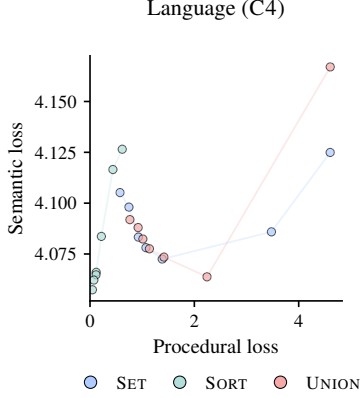

*Figure 18.* Transferability of procedural pretraining. Relationship between procedural validation loss and downstream loss on C4. For SET and UNION, transfer is strongest at intermediate procedural losses, with both underfitting and overfitting harming generalization. For SORT, continually decreasing procedural loss consistently improves transfer.

## J. The Effect of Vocabulary Size

We investigate the effect of vocabulary size during procedural pretraining.

**Setup.** Models are pretrained on SET, SORT, and UNION with vocabularies from 25 to 500 symbols (the main results use 100 by default), then transferred to WIKITEXT using full-model transfer. Evaluation perplexity is reported after fine-tuning.

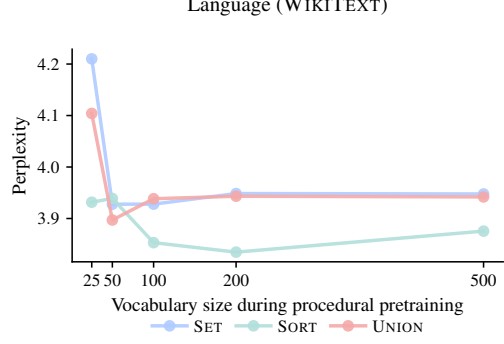

*Figure 19.* Effect of vocabulary size during procedural pretraining on WIKITEXT. Small vocabularies (25–50) degrade transfer performance, while moderate sizes (∼100-200) are sufficient. Larger vocabularies offer no further improvement.

**Results.** As shown in Figure 19, very small vocabularies (25–50) harm transfer, leading to higher perplexity. For SET and UNION, performance stabilizes once the vocabulary reaches a moderate size (∼100), with larger sizes offering no further gains. SORT benefits modestly at 200 but declines at 500. Overall, procedural pretraining is most effective within a moderate

vocabulary range, too small harms transfer, while too large brings no improvement or negative return.

## K. Weight Decay Ablation

In the main paper, natural language experiments use a weight decay of 0.1 during procedural pretraining, following Hu et al. (2025). To test this choice, we reduce the weight decay to 0.01 (the value used for code and math) and evaluate performance on C4 semantic pretraining. The takeaway that MLP-only transfer is best for natural language remains unchanged, showing that our findings are robust to this hyperparameter.

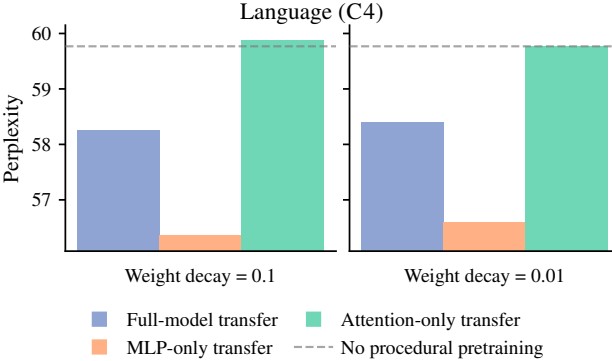

*Figure 20.* Effect of weight decay during procedural pretraining on C4. Changing weight decay from 0.1 to 0.01 does not alter the outcome: MLP-only transfer remains the best configuration for natural language.

## L. Scaling Procedural Pretraining

Extending the findings of Section 5.2, we scale both model size and semantic pretraining data size. We increase the architecture to 350M parameters, and further to a 1.3B-parameter model (architectural hyperparameters follow Biderman et al. (2023)), while scaling natural-language pretraining to 1.6B / 6.6B C4 tokens and 4.8B / 10.5B CODEPARROT tokens respectively.

For 350M models and 1.3B models, we use a learning rate of $3 \times 10^{-4}$ and $2 \times 10^{-4}$, following Biderman et al. (2023). We also use larger batch sizes and/or training steps for the semantic pretraining to increase the semantic tokens. Other hyperparameters follow Appendix E. We utilise UNION for procedural pretraining on both C4 and CODEPARROT.

**Additive setting.** We find procedurally pretrained models continue to substantially outperform their non-procedural counterparts across all scales (Table 3). This shows that the benefits of procedural pretraining persist at substantially larger scales in both model capacity and dataset size.

| Model | C4 (Perplexity ↓) | CODEPARROT (Perplexity ↓) |
|---|---|---|
| **350M parameters** | | |
| NO PROCEDURAL PRETRAINING | 40.3 | 4.97 |
| OURS (UNION) | **39.0** | **4.62** |
| **1.3B parameters** | | |
| NO PROCEDURAL PRETRAINING | 28.8 | 3.45 |
| OURS (UNION) | **27.3** | **3.36** |

*Table 3.* Perplexity of language models with and without procedural pretraining at increased scale. 350M-parameter models are pretrained on 1.6B C4 tokens and 4.8B CODEPARROT tokens. 1.3B-parameter models are pretrained on 6.6B C4 tokens and 10.5B CODEPARROT tokens. **Procedural pretraining consistently improves perplexity across both scale regimes.**

We additionally report BLiMP evaluation for the larger C4-trained models. These show that procedural pretraining imparts lasting gains in syntactic and morphological generalization at a larger scale (Table 4).

| Model | BLiMP (Accuracy ↑) |
|---|---|
| **350M parameters** | |
| NO PROCEDURAL PRETRAINING | 71.5 |
| OURS (UNION) | **72.9** |
| **1.3B parameters** | |
| NO PROCEDURAL PRETRAINING | 73.2 |
| OURS (UNION) | **75.5** |

*Table 4.* BLiMP accuracy for language models with and without procedural pretraining at increased scale. **Procedural pretraining consistently improves grammatical acceptability across both scales.**

**Substitutive setting.** We further evaluate the substitutive setting at the 1.3B-parameter scale. Specifically, we use only 82M procedural tokens. Despite this minimal additional data, procedural pretraining enables the model to match baseline performance using just 66% of the C4 data and 75% of the CODEPARROT data. This corresponds to a reduction of 2.1B C4 tokens and 2.5B CODEPARROT tokens in semantic pretraining.

## M. Results on downstream tasks

This section provides the extended downstream fine-tuning or few-shot results referenced in Section 5.2. See Appendix E.4 for additional experimental details.

### M.1. Finetuning Results

**Setup.** To investigate whether the benefits of procedural pretraining persist after downstream fine-tuning, we conduct an additional fine-tuning step. Specifically, we finetune the language models (pretrained on C4) on both WIKITEXT-103 and GLUE tasks independently. The code models (pretrained on CODEPARROT) are finetuned and evaluated on PY150. For WikiText-103, we use the SORT model, as it obtains the lowest perplexity on C4. For GLUE and PY150, we instead use the UNION model as it has demonstrated consistently strong performance across a broad range of downstream tasks.

**Results.** Consistent with the main findings of enhancing semantic pretraining, the procedurally pretrained models continue to outperform the baseline across these downstream tasks (Table 5 and Table 6). This shows that the benefits of procedural data persist after fine-tuning on downstream tasks, suggesting the potential of using procedural pretraining to enhance the practical utility of models.

| Model | WIKITEXT-103 (Perplexity ↓) | PY150 (Accuracy ↑) |
|---|---|---|
| NO PROCEDURAL PRETRAINING | 33.0 | 60.5 |
| OURS | **32.3** | **62.1** |

*Table 5.* Downstream fine-tuning results on WIKITEXT-103 (perplexity; after C4 pretraining) and PY150 (accuracy; after CODEPARROT pretraining), comparing models with and without procedural pretraining.

| | COLA | SST-2 | MRPC | QQP | STS-B | MNLI | QNLI | RTE | WNLI | **Avg** |
|---|---|---|---|---|---|---|---|---|---|---|
| NO PROC. P.T. | 69.1 | 85.3 | 70.8 | 84.3 | 55.1 | 72.1 | 79.9 | 57.4 | 42.3 | 68.5 |
| OURS | 68.9 | 87.6 | 69.6 | 84.8 | 68.8 | 72.7 | 81.3 | 55.6 | 52.1 | **71.3** |

*Table 6.* GLUE scores after C4 pretraining, comparing the baseline without procedural pretraining to our model with procedural pretraining.

*Table 7.* Zero-shot commonsense reasoning results on ARC-Easy and HellaSwag, comparing models with and without procedural pretraining.

| Model | ARC-Easy (Accuracy ↑) | HellaSwag (Accuracy ↑) |
|---|---|---|
| NO PROCEDURAL PRETRAINING | 37.3 | 27.6 |
| OURS | **38.2** | **28.3** |

### M.2. Zero-shot Results

**Setup.** We further evaluate whether the benefits of procedural pretraining persist in zero-shot downstream evaluation. We compare models with and without procedural pretraining on commonsense reasoning benchmarks, using ARC-Easy (Clark et al., 2018), HellaSwag (Zellers et al., 2019).

**Results.** Consistent with the fine-tuning results in Appendix M.1, procedural pretraining improves performance on both commonsense reasoning benchmarks (Table 7). These results indicate that the gains from procedural data also transfer to zero-shot commonsense reasoning.

## N. Additional Results

### N.1. Algorithmic Reasoning Tasks

| Pretraining task | HAYSTACK | ADDITION | REVERSED ADDITION | MULTIPLICATION | SORTING |
|---|---|---|---|---|---|
| RAND INIT. | $11.3 \pm 0.4$ | $59.1 \pm 7.0$ | $76.4 \pm 23.2$ | $42.7 \pm 5.3$ | $82.7 \pm 11.6$ |
| 4-DYCK | $98.3 \pm 1.1$ | $52.7 \pm 0.3$ | $35.7 \pm 2.5$ | $46.7 \pm 4.6$ | $56.3 \pm 19.2$ |
| 8-DYCK | $93.6 \pm 1.3$ | $53.4 \pm 0.3$ | $48.9 \pm 4.9$ | $44.5 \pm 0.9$ | $98.7 \pm 0.3$ |
| 16-DYCK | $96.9 \pm 1.0$ | $87.8 \pm 4.2$ | $83.5 \pm 0.6$ | $39.4 \pm 3.3$ | $95.5 \pm 1.0$ |
| 4-DYCK SHUFFLE | $7.3 \pm 0.6$ | $54.5 \pm 0.2$ | $87.8 \pm 12.9$ | $41.8 \pm 3.7$ | $61.0 \pm 1.4$ |
| 8-DYCK SHUFFLE | $9.6 \pm 0.3$ | $67.7 \pm 0.8$ | $90.1 \pm 5.9$ | $37.4 \pm 0.1$ | $84.1 \pm 5.7$ |
| 16-DYCK SHUFFLE | $18.6 \pm 26.3$ | $70.8 \pm 5.5$ | $87.0 \pm 12.8$ | $44.0 \pm 0.1$ | $71.1 \pm 5.4$ |
| STACK | $55.2 \pm 39.3$ | $62.3 \pm 5.3$ | $34.9 \pm 0.2$ | $46.6 \pm 2.0$ | $21.3 \pm 0.6$ |
| IDENTITY | $18.8 \pm 14.3$ | $54.7 \pm 0.2$ | $42.7 \pm 0.9$ | $46.6 \pm 2.7$ | $19.9 \pm 0.5$ |
| SET | $18.9 \pm 26.6$ | $53.4 \pm 0.1$ | $44.6 \pm 5.1$ | $43.5 \pm 8.4$ | $93.5 \pm 1.6$ |
| UNION | $9.8 \pm 1.1$ | $48.6 \pm 0.7$ | $50.8 \pm 0.2$ | $63.5 \pm 2.3$ | $16.9 \pm 0.5$ |
| REVERSE | $33.3 \pm 22.4$ | $46.1 \pm 2.3$ | $46.8 \pm 1.33$ | $54.4 \pm 3.2$ | $16.7 \pm 0.5$ |
| DELETE | $52.6 \pm 22.4$ | $60.7 \pm 4.19$ | $40.0 \pm 1.8$ | $61.9 \pm 1.4$ | $20.1 \pm 0.6$ |
| ECA RULE 110 | $10.5 \pm 0.5$ | $69.6 \pm 7.9$ | $91.1 \pm 16.1$ | — | $76.9 \pm 1.4$ |
| BEST MODEL SHUFFLED | $10.3 \pm 0.5$ | $52.0 \pm 0.3$ | $65.0 \pm 21.4$ | $48.4 \pm 4.4$ | $69.9 \pm 2.2$ |

*Table 8.* Full results across all pretraining tasks and algorithmic reasoning tasks. Each cell reports the mean accuracy $\pm$ standard deviation over 10 random seeds, except for MULTIPLICATION, which is over 3 seeds. The means of these results are visualised in Figure 2.

| Pretraining task | FULL TRANSFER | MLP ONLY | ATTENTION ONLY |
|---|---|---|---|
| 4-DYCK | $98.3 \pm 1.1$ | $8.7 \pm 0.5$ | $11.6 \pm 0.5$ |
| 16-DYCK SHUFFLE | $18.6 \pm 26.3$ | $8.9 \pm 0.9$ | $16.5 \pm 10.6$ |
| STACK | $55.2 \pm 39.3$ | $7.1 \pm 0.6$ | $98.9 \pm 0.8$ |
| IDENTITY | $18.8 \pm 14.3$ | $7.0 \pm 0.9$ | $99.0 \pm 1.7$ |
| SET | $18.9 \pm 26.6$ | $8.3 \pm 0.7$ | $88.9 \pm 27.1$ |
| UNION | $9.8 \pm 1.1$ | $8.2 \pm 0.7$ | $11.7 \pm 0.4$ |
| REVERSE | $33.3 \pm 22.4$ | $7.3 \pm 1.2$ | $98.6 \pm 0.8$ |
| DELETE | $52.6 \pm 22.4$ | $8.4 \pm 0.8$ | $91.8 \pm 3.5$ |
| ECA | $10.5 \pm 0.5$ | $8.7 \pm 1.0$ | $11.6 \pm 1.0$ |

*Table 9.* HAYSTACK task accuracy (mean $\pm$ standard deviation over 10 seeds) for models initialized with weights from different pretraining tasks. We report results for full model transfer, MLP-transfer, and attention-transfer.

| Pretraining task | FULL TRANSFER | MLP ONLY | ATTENTION ONLY |
|---|---|---|---|
| 16-DYCK | $87.8 \pm 4.2$ | $60.0 \pm 6.6$ | $59.2 \pm 10.4$ |
| 16-DYCK SHUFFLE | $70.8 \pm 5.5$ | $61.7 \pm 6.9$ | $55.3 \pm 4.9$ |
| STACK | $62.3 \pm 5.3$ | $61.1 \pm 9.4$ | $56.2 \pm 5.0$ |
| IDENTITY | $54.7 \pm 0.2$ | $58.3 \pm 7.2$ | $69.7 \pm 13.1$ |
| SET | $53.4 \pm 0.1$ | $59.6 \pm 6.4$ | $81.1 \pm 12.2$ |
| UNION | $48.6 \pm 0.7$ | $65.0 \pm 12.2$ | $59.8 \pm 9.0$ |
| REVERSE | $46.1 \pm 2.3$ | $57.8 \pm 7.0$ | $60.9 \pm 7.9$ |
| DELETE | $60.7 \pm 4.2$ | $59.2 \pm 8.1$ | $63.3 \pm 14.0$ |
| ECA | $69.6 \pm 7.9$ | $63.1 \pm 14.4$ | $65.8 \pm 12.8$ |

*Table 10.* ADDITION task accuracy (mean $\pm$ standard deviation over 10 seeds) for models initialized with weights from different pretraining tasks. We report results for full model transfer, MLP-transfer, and attention-transfer.

| Pretraining task | FULL TRANSFER | MLP ONLY | ATTENTION ONLY |
|---|---|---|---|
| 16-DYCK | $83.5 \pm 0.6$ | $64.0 \pm 26.4$ | $49.1 \pm 20.3$ |
| 8-DYCK SHUFFLE | $90.1 \pm 5.9$ | $65.8 \pm 24.8$ | $63.3 \pm 18.1$ |
| STACK | $34.9 \pm 0.2$ | $74.4 \pm 24.7$ | $42.1 \pm 8.1$ |
| IDENTITY | $42.7 \pm 0.9$ | $71.7 \pm 29.2$ | $45.2 \pm 3.7$ |
| SET | $44.6 \pm 5.1$ | $71.2 \pm 23.7$ | $54.4 \pm 10.4$ |
| UNION | $50.8 \pm 0.2$ | $72.3 \pm 29.6$ | $50.3 \pm 16.5$ |
| REVERSE | $46.8 \pm 1.3$ | $75.8 \pm 27.1$ | $44.6 \pm 3.4$ |
| DELETE | $40.0 \pm 1.8$ | $55.2 \pm 23.0$ | $44.6 \pm 9.2$ |
| ECA | $91.1 \pm 16.1$ | $70.5 \pm 31.6$ | $75.5 \pm 27.2$ |

*Table 11.* REVERSED ADDITION task accuracy (mean $\pm$ standard deviation over 10 seeds) for models initialized with weights from different pretraining tasks. We report results for full model transfer, MLP-transfer, and attention-transfer.

| Pretraining task | FULL TRANSFER | MLP ONLY | ATTENTION ONLY |
|---|---|---|---|
| 8-DYCK | 98.7±0.3 | 72.8±3.1 | 71.4±5.7 |
| 8-DYCK SHUFFLE | 84.1±5.7 | 78.2±8.6 | 62.9±6.7 |
| STACK | 21.3±0.6 | 71.0±2.2 | 77.5±12.2 |
| IDENTITY | 19.9±0.5 | 74.5±8.1 | 91.3±10.1 |
| SET | 93.5±1.6 | 73.5±1.5 | 98.1±2.8 |
| UNION | 16.9 ± 0.5 | 72.3 ± 1.9 | 76.4 ± 16.4 |
| REVERSE | 16.7 ± 0.5 | 71.2 ± 2.6 | 82.1 ± 15.1 |
| DELETE | 20.1 ± 0.6 | 78.0 ± 10.9 | 81.3 ± 24.3 |
| ECA | 76.9 ± 1.4 | 77.1±8.1 | 73.9±3.2 |

*Table 12.* SORTING task accuracy (mean ± standard deviation over 10 seeds) for models initialized with weights from different pretraining tasks. We report results for full model transfer, MLP-transfer, and attention-transfer.

| Perturbation | HAYSTACK | ADDITION | REVERSED ADDITION | SORTING |
|---|---|---|---|---|
| Pretrained | 98.9 ± 0.8 | 87.8 ± 4.2 | 90.1 ± 5.9 | 98.7 ± 0.3 |
| Shuffled | 17.2 ± 12.7 | 61.0 ± 9.1 | 82.9 ± 23.5 | 94.2 ± 4.2 |
| 0.01 noise | 98.6 ± 1.7 | 77.6 ± 20.1 | 74.0 ± 21.0 | 96.0 ± 7.6 |
| 0.05 noise | 50.8 ± 30.5 | 62.1 ± 13.3 | 91.0 ± 15.7 | 71.9 ± 26.1 |
| 0.10 noise | 32.9 ± 6.1 | 56.4 ± 7.4 | 83.6 ± 21.5 | 37.9 ± 5.8 |
| Random init | 11.3 ± 0.4 | 59.1 ± 7.0 | 76.4 ± 23.2 | 82.7 ± 11.6 |

*Table 13.* Mean accuracy (± standard deviation over 10 seeds) across five algorithmic tasks under different perturbation conditions. Pretrained models were selected based on best individual performance per task: STACK (attention-transfer) for HAYSTACK, 16-DYCK for ADDITION (full-transfer), 8-DYCK SHUFFLE for REVERSED ADDITION (full-transfer), 8-DYCK for SORTING (full-transfer).

## N.2. Semantic Data

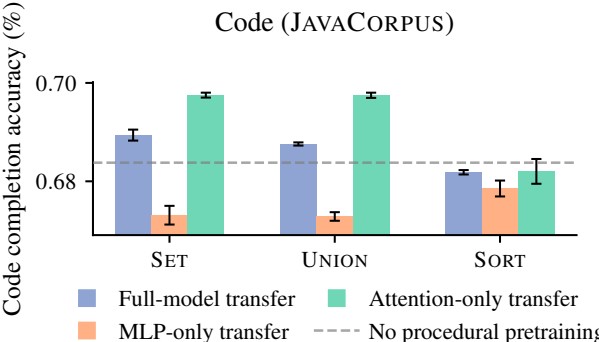

*Figure 21.* Token level code completion accuracy on JAVACORPUS from (Lu et al., 2021). We compare partial transfer of pretrained weights with full-model transfer. This extends the partial transfer analysis from Figure 6 in the main paper, showing Attention-only transfer is superior for code in isolation.

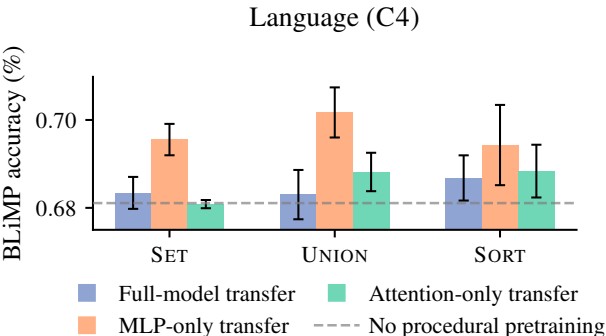

*Figure 22.* BLiMP accuracy (Warstadt et al., 2020) after training on C4. We compare partial transfer of pretrained weights with full-model transfer. Consistent with Figure 6, MLP-only transfer achieves the best performance on grammatical understanding.

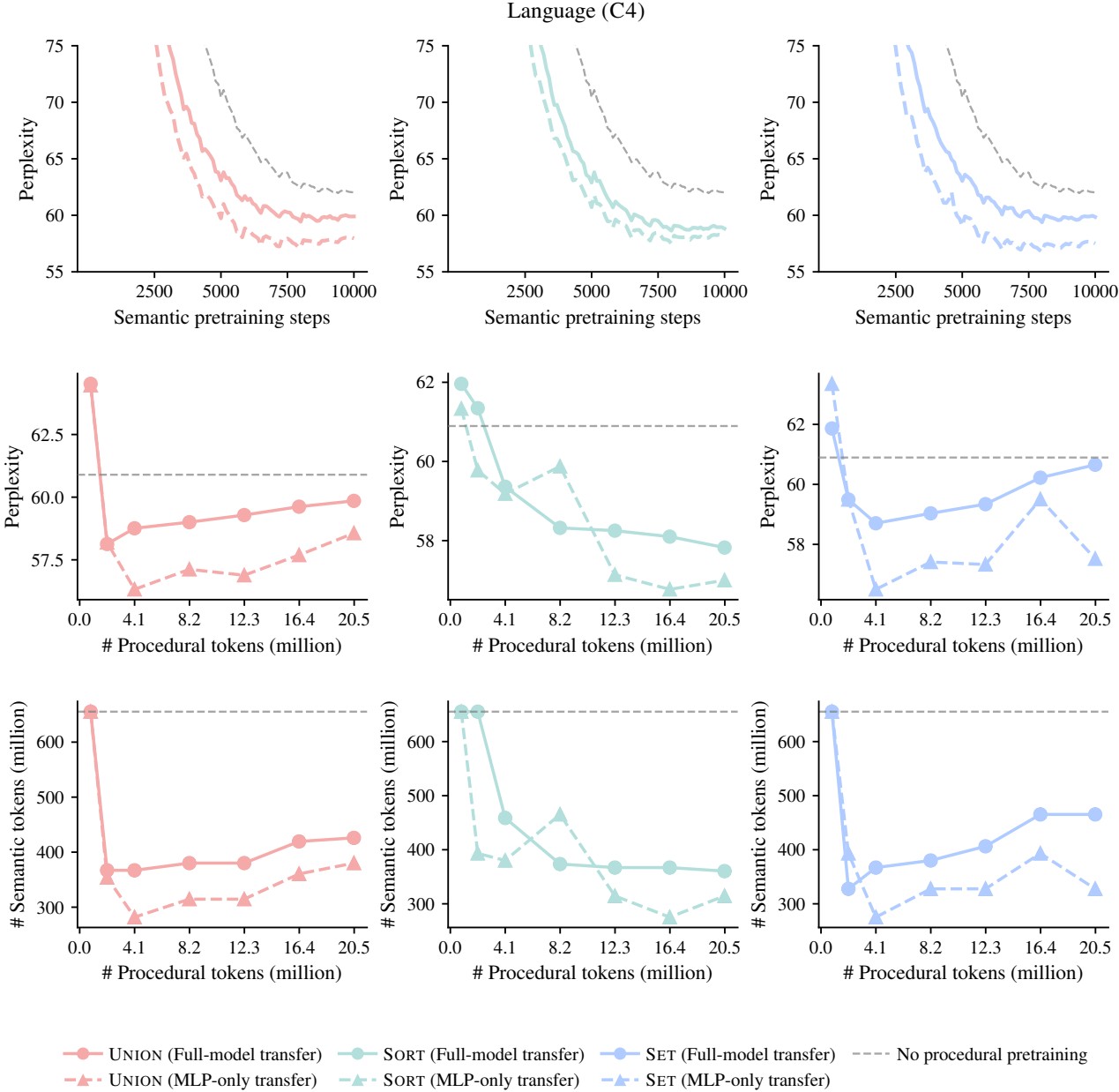

*Figure 23.* Comparison of MLP-only transfer and full-model transfer on C4 for UNION, SORT and SET. (**Top**) Perplexity curves during semantic pretraining. (**Middle**) Additive setting results. (**Bottom**) Substitutive setting results. Across all views, MLP-only transfer outperforms full transfer, confirming that procedurally pretrained MLP layers are especially effective for natural language.

## N.3. Weight Mixture

| | HAYSTACK | ADDITION | REVERSED ADDITION | SORT |
|---|---|---|---|---|
| No procedural pretraining | $11.3_{\pm 0.4}$ | $59.1_{\pm 7.0}$ | $76.4_{\pm 23.2}$ | $82.7_{\pm 11.6}$ |
| SET  (full-model transfer) | $18.9_{\pm 26.6}$ | $53.4_{\pm 0.1}$ | $44.6_{\pm 5.1}$ | $93.5_{\pm 1.6}$ |
| SET  (attention-only transfer) | $88.9_{\pm 27.1}$ | $\mathbf{81.1}_{\pm 12.2}$ | $54.4_{\pm 10.4}$ | $98.1_{\pm 2.8}$ |
| ECA  (full-model transfer) | $10.5_{\pm 0.5}$ | $69.6_{\pm 7.9}$ | $\mathbf{91.0}_{\pm 16.1}$ | $76.9_{\pm 1.4}$ |
| ECA  (MLP-only transfer) | $8.71_{\pm 1.0}$ | $63.1_{\pm 14.4}$ | $70.5_{\pm 31.6}$ | $77.1_{\pm 8.1}$ |
| SET (attention) + ECA (MLP) | $\mathbf{94.4}_{\pm 2.5}$ | $\underline{80.3}_{\pm 13.9}$ | $\underline{82.9}_{\pm 16.9}$ | $\mathbf{99.4}_{\pm 0.2}$ |

*Table 14.* **Pretrained models combined at the weight level.** We combine SET-pretrained attention layers with ECA-pretrained MLPs (last row). This yields the strong performance across all four tasks, whereas single-source models show weaknesses in at least one task.

