# OpenReview forum: "Procedural Pretraining: Warming Up Language Models with Abstract Data"
_ICML.cc/2026/Conference — ICML 2026 spotlight_

### Official Review · Reviewer_ci8x · 2026-03-10

**Soundness:** 2
**Presentation:** 3
**Significance:** 2
**Originality:** 3
**Overall Recommendation:** 3
**Confidence:** 4

**Summary:**

This paper introduces procedural pretraining, a paradigm in which LLMs are pretrained with procedural data before training on general copora. Procedural data are generated by formal langauges and have a clearer logical structure than natural text. The authors show that warming up with such data significantly speeds up convergence on natural text and improves downstream performance.

**Compliance With Llm Reviewing Policy:**

Affirmed.

**Final Justification:**

I thank the authors for carefully addressing my concerns, particularly about W1. While the proposed procedural pretraining paradigm is intriguing, the execution of experimental validation fundamentally falls short. To summarize my remaining concerns:
- Confounding experiment setups. The authors pretrain on multiple datasets individually for different downstream evaluations, and each experiment uses a different backbone model config. A proper pretrain setup, instead, would be to train on a single bulk corpus (such as FineWeb or C4), then evaluate diverse downstream tasks.
- Non-standard pretrain protocol. I think the experiment setup could be closer to recent industry standards, to name a few:
    - Warmup-Stable-Decay LR scheduler, which has reported better potential than cosine schedulers.
    - BFloat16 training. The authors adopted the Pythia suite whose configurations (such as lr) are still tuned for fp16.
    - Broad evaluation. Evaluation should be based on at least around 10 different benchmarks (as mentioned in W2), to ensure completeness and robustness. Otherwise, observed gains on a few selected downstream tasks could be accomplished by sacrificing others.
- Insufficient training horizon. Applicability to modern pretraining highly depends on the diminishing gains in the long run. While training on trillions of tokens is prohibitively expensive, the authors should at least train on optimal compute for each experiment. Moreover, gains at smaller scales might not even carry to larger scales.
- Untuned or potentially suboptimal training hyperparameters. The experiments rely on the Pythia suite, whose training setup differs from current mainstream LLM pretraining practices. Aligning with more recent training frameworks (e.g., nanochat) or modern training conventions as stated above would improve the relevance of the results to current LLM research.

Based on the above limitations, I will increase my score from 2 to 3.

**Key Questions For Authors:**

See weaknesses.

**Limitations:**

yes

**Strengths And Weaknesses:**

### Strengths
1. Multiple pretraining datasets are tested.
2. Discussions on various procedural data types is intriguing.

### Weaknesses
1. Missing important baseline. The proposed method follows: (i) procedural pretraining with small batch size (in terms of tokens per batch) and large update steps (ii) followed by semantic pretraining with large batch size and fewer update steps. As a baseline, authors should test: (i) semantic pretraining with the same batch size and update steps as procedural pretraining (ii) followed by regular semantic pretraining with large batch size. This baseline helps **isolate the advantage of more gradient update steps from the benefits of procedural data alone**. A similar configuration the authors should also test is: (i) procedural pretraining with the same batch size as semantic pretraining (ii) followed by regular semantic pretraining. This config is then compared with full semantic pretraining with the same amount of update steps and token count, to verify the effectiveness of procedural pretraining with fewer update steps and larger batch size.
2. Lack of evaluation on general reasoning benchmarks, such as PIQA, SIQA, Winogrande, ARC-Easy, HellaSwag, MMLU, etc. These are common benchmarks used for evaluating pretrained language models. I would advice showing how procedural pretraining affects performance on these tasks.
3. The model and data scales are too small to be reliably meaningful, and a majority of experiments are undertrained by today's standards (e.g. according to Chinchilla scaling law). For the largest scale experiment in your paper, I recommend a minimum of 1B model with 20B semantic tokens.
4. Need more details about learning rate tuning. For example, training on C4 uses LR $5\times10^{-4}$ with batch size $32\times2048$. From my experience, a model with 124M parameters can safely train with a learning rate of $2\times10^{-3}$, while an 1B model would train stably with $6\times10^{-4}$.

---

> ### Author Rebuttal · Authors · 2026-03-30
>
> Thanks for the review. We are glad that the experiments were deemed "*comprehensive*" and the discussion "*intriguing*".
> We address some **misunderstandings** (W1) and provide **new additional results** (W1&2) that we are also adding to the paper.
>
> ---
>
> **`W1: Missing baseline that matches batch size and gradient update steps`**
>
> > *The proposed method follows: (i) procedural pretraining with small batch size (tokens per batch) and large update steps (ii) followed by semantic pretraining with large batch size and fewer steps*
>
> It's actually the opposite: *semantic* pretraining has much larger steps, hence it's unlikely that the improvements are due to hyperparameters alone.
> Indeed, procedural pretraining uses both much fewer steps (100-2,500 vs. 10,000-50,000) and much fewer tokens per step (batch size 64 vs. 32-64 and smaller sequences, 8-64 vs. 512-2048). This is why procedural pretraining is tremendously cheaper than its semantic counterpart.
>
> Regarding the suggested baseline, current results in the paper already suggest superiority to it. **Using procedural training (only 100-2500 steps and much fewer tokens per step) can save 4500/6800/7000 gradient steps of semantic pretraining on langage/code/math respectively, while still achieving the same performance.** This means a "semantic warmup" baseline (suggested by the reviewer) with the same number of steps as procedural pretraining could not catch up.
>
> Nonetheless, **we performed the additional requested baseline** with a "warmup" on semantic data (matching the number of steps, batch size, and sequence length). Our method indeed performs significantly better than this baseline.
>
> |   | C4 $\downarrow$ | CodeParrot $\downarrow$ |
> |---|---|---|
> | Semantic warmup baseline | 35.30  | 5.15  |
> | Procedural warmup (ours)  | **33.21** |  **4.84** |
>
>
> These results use a 124M model/2B token (Chinchilla scaling). We are also running larger models (350M/7B) and will update these results if available in time.
>
> ㅤ
>
> ---
>
> **`W2: Lack of evaluation on general reasoning benchmarks`**
>
> **We performed additional experiments and evaluations as requested**. We first report 3-shot pass@k accuracy (%) on the MBPP benchmark (*Program Synthesis with Large Language Models*, Austin et al. 2021) using models trained on CodeParrot from Figure 5.
> The baseline model completely fails to solve any MBPP problem even with 80 attempts per problem, while our procedurally-pretrained model starts showing non-random capabilities.
> The compared models are identical in all respects except for the procedural warm-up, which only requires a trivial amount of extra compute.
>
> | Pass@k | 1 | 5 | 10 | 80 (MBPP paper's setup) |
> |---|---|---|---| --- |
> | W/o procedural pretraining (baseline) | 0.0  | 0.0  |  0.0  | 0.0 |
> | W/ procedural pretraining (ours)  | **0.2** |  **0.6** |  **1.0**  | **3.4** |
> |
>
> We also performed a **new additional evaluation on ARC-Easy and HellaSwag** (zero-shot accuracy in %) using the C4-trained models.
> The procedurally-pretrained model shows a clear improvement compared to the baseline.
>
> |   | ARC-Easy | HellaSwag |
> |---|---|---|
> | W/o procedural pretraining (baseline) | 37.3  | 27.6  |
> | W/ procedural pretraining (ours)  | **38.2** |  **28.3** |
> |
>
> ㅤ
>
> ---
>
> **`W3: Model and data scales are too small`**
>
> Our experiments are already much larger (1.3B parameters, 10.5B tokens) than in prior work on procedural data ([1], an "*outstanding paper*" at ACL 2025, uses a 1B model with merely 1.6B tokens).
>
> **Additionally, we performed new experiments as requested** following Chinchilla scaling (20 tokens/parameter) reported above while addressing `W1`.
>
> There is obviously room for further validation at *industrial scales*, which we clearly acknowledge in the discussion section. We focus instead on scientific contributions including (1) studying the mechanisms behind diverse procedural data, (2) showing procedural data helps pretraining on diverse domains across a range of model scales, (3) laying out important next steps including a proof-of-concept for combining multiple types of procedural data.
>
> [1] *Between Circuits and Chomsky: Pre-pretraining on Formal Languages Imparts Linguistic Biases*, Hu et al. 2025
>
> ㅤ
>
> ---
>
> **`W4: Need more details about learning rate tuning`**
>
> This is mentioned in Appendix E.3. We follow the settings from reputable prior studies.
> - For 124M models on C4 and CodeParrot, we follow the training recipe from **Huggingface's CodeParrot-small** and ***Between Circuits and Chomsky paper*** (Hu et al. 2025). For DeepMind-Math, we also follow the settings from **DeepMind's original paper** (Saxton et al. 2019).
> - For 350M and 1.3B models, we follow the architectures and learning rates from **EleutherAI's Pythia** (Biderman et al. 2023).
>
> ㅤ
>
> ---
>
> Thanks again for these suggestions. We are updating the paper accordingly and believe that it has significantly improved as a result. Let us know if there is any remaining concern!

---

> > ### Author Rebuttal · Reviewer_ci8x · 2026-04-03
> >
> > I thank the authors for carefully addressing my concerns, particularly about W1. While the proposed procedural pretraining paradigm is intriguing, the execution of experimental validation fundamentally falls short. To summarize my remaining concerns:
> > - Confounding experiment setups. The authors pretrain on multiple datasets individually for different downstream evaluations, and each experiment uses a different backbone model config. A proper pretrain setup, instead, would be to train on a single bulk corpus (such as FineWeb or C4), then evaluate diverse downstream tasks.
> > - Non-standard pretrain protocol. I think the experiment setup could be closer to recent industry standards, to name a few:
> >     - Warmup-Stable-Decay LR scheduler, which has reported better potential than cosine schedulers.
> >     - BFloat16 training. The authors adopted the Pythia suite whose configurations (such as lr) are still tuned for fp16.
> >     - Broad evaluation. Evaluation should be based on at least around 10 different benchmarks (as mentioned in W2), to ensure completeness and robustness. Otherwise, observed gains on a few selected downstream tasks could be accomplished by sacrificing others.
> > - Insufficient training horizon. Applicability to modern pretraining highly depends on the diminishing gains in the long run. While training on trillions of tokens is prohibitively expensive, the authors should at least train on optimal compute for each experiment. Moreover, gains at smaller scales might not even carry to larger scales.
> > - Untuned or potentially suboptimal training hyperparameters. The experiments rely on the Pythia suite, whose training setup differs from current mainstream LLM pretraining practices. Aligning with more recent training frameworks (e.g., nanochat) or modern training conventions as stated above would improve the relevance of the results to current LLM research.
> >
> > Based on the above limitations, I will increase my score from 2 to 3.

---

> > > ### Author Response · Authors · 2026-04-03
> > >
> > > Thank you for your continued engagement and thoughtful comments. We address the remaining concerns as follows.
> > >
> > > **`Confounding experiment setups`**
> > >
> > > We respectfully disagree that this is a weakness. In fact, the experimental setup is controlled and constitutes one of the strengths of our work. We believe there may some misunderstanding about the experimental setup.
> > >
> > > - Our experiments **already follow** the suggested "single bulk corpus" paradigm (C4 for language, CodeParrot for code), as suggested by the reviewer for a "*proper pretraining setup*". These are both standard sources used respectiveöy as **sole pretraining corpora** for language models. CodeParrot was for example *the only source* for training a 1.5B coding model by HuggingFace.
> > > - We further consider fine-grained domains such as the math portion of The Pile (the "*DeepMind-Math*" dataset) to study detailed questions such as: do different types of procedural data generalize equally well across language, code, and math domains? Where is the useful information localized for each domain? **These questions cannot be answered with a single pretraining source.**
> > >
> > > - > “*Each experiment uses a different backbone model config*”
> > >
> > >
> > >   This is **not** correct. We use the exact same backbone configuration within each set of experiments for a fair comparison between models with and without procedural pretraining, and we also show that our results generalize across different backbone configurations (e.g., model sizes).
> > >
> > >
> > > **`Untuned or potentially suboptimal training hyperparameters / Non-standard pretrain protocol`**
> > >
> > > - Our experiments follow strong prior work, as noted in the rebuttal. This setup is intentionally chosen to **isolate the effects of procedural pretraining**, rather than optimizing absolute performance using extensive training tricks. Evaluating procedural pretraining across a wide range of modern LLM optimization techniques and training recipes is beyond the scope of this (already extensive) work.
> > >
> > > - > “*The experiment setup could be closer to recent industry standards*”:
> > >
> > >   This paper is **not** about scaling up to industrial-scale training, but about **understanding the benefits and mechanisms of procedural data**. In particular, we answer the following questions.
> > >   - Can procedural data (which is non-semantic) help semantic pretraining across diverse domains?
> > >   - What specific capabilities is procedural data inducing or reinforcing in the model?
> > >   - Are various types of procedural data equally helpful? Are certain types mroe useful for certain domains?
> > >   - What are the next steps towards a widespread applicability and adoptions of procedural data for training LLMs? We specifically propose and show that optimizing mixtures of procedural data is a promising step.
> > >
> > >   The paper **addresses all of these questions, which are necessary steps towards an industrial adoption up of procedural pretraining**. Our findings suggest that scaling is particularly promising. An industrial-scale study (e.g., ≥3B models trained on trillions of tokens, where many LLM benchmarks only begin to exhibit non-random accuracy) would warrant a whole separate paper (and significantly more resources). We explicitly acknowledge this in the discussion section.
> > >
> > > **`Insufficient training horizon`**
> > >
> > > As mentioned in our rebuttal, our experiments are already substantially larger than prior published work on this topic (Hu et al., outstanding paper at ACL 2025). And **we already have conducted new experiments on the reviewer's request** following Chinchilla scaling (20 tokens/parameter) where **the benefits of procedural pretraining clearly do persist**.

---

### Official Review · Reviewer_87jz · 2026-03-12

**Soundness:** 4
**Presentation:** 4
**Significance:** 4
**Originality:** 3
**Overall Recommendation:** 5
**Confidence:** 4

**Summary:**

This paper argues that a small amount of procedural pretraining on abstract algorithmic tasks, such as set operations and sequence reversal, can improve downstream performance and perplexity after standard pretraining. The authors first show that different types of procedural data strengthen different algorithmic abilities, and that these abilities appear to be localized in different parts of the model, such as attention layers or MLPs. They then show that procedural pretraining transfers to more realistic domains, including natural language, code, and math, even with only a small number of additional procedural pretraining tokens. In a substitutive setting, they further suggest that procedural pretraining can be more data-efficient than standard pretraining alone, achieving similar perplexity with less semantic training data and fewer overall training tokens. They also find that attention layers are especially important for transferring code-related skills, whereas MLPs matter more for natural language. Finally, they explore two ways of combining procedural skills via pretraining on mixtures of procedural data and surgically merging weights from separately pretrained models, and report promising results from both. This complementary view of procedural pretraining differs from prior work.

**Compliance With Llm Reviewing Policy:**

Affirmed.

**Final Justification:**

My final score for this paper is accept. I believe that the authors addressed all of my concerns. After reviewing the concerns from other reviewers about the non-standard training protocol and range of evaluations, I do believe that the range of experiments for this paper could have been more expansive and the evaluations more thorough to support the claims. I still believe that this paper should be accepted, as it presents a well-written and strong contribution to the topic of procedural pre-training.

**Key Questions For Authors:**

1. In Section 4.1, can the authors provide more intuition for why particular procedural pretraining tasks transfer to particular downstream abilities? For example, why does K-Dyck improve Haystack, and what properties of ECA Rule 110 make it helpful for the tasks where it shows gains?

2. In Section 4.2, can the authors expand on the mechanistic interpretation of transfer across model components? More specifically, under what conditions are transferred skills primarily stored in FFNs/MLPs, and when is attention the main component enabling transfer?

**Limitations:**

Yes

**Strengths And Weaknesses:**

## Soundness
This is a very solid paper overall, and I do not see meaningful weaknesses in terms of soundness. The experimental section is comprehensive, and the claims are supported by a wide range of results. The controls are also strong; in particular, I appreciated the shuffled-procedural-data baselines, which preserve token distributions while disrupting structure and therefore help isolate the effect of procedural structure itself.

My only minor criticism is that I would have liked more discussion of why certain procedural pretraining tasks help specific downstream abilities. For example, in Section 4.1, it would be useful to better understand why K-Dyck improves Haystack, or why ECA Rule 110 seems to help particular tasks. Similarly, in Section 4.2, I would have liked more interpretation of when information is primarily stored in FFNs versus when attention is the main mechanism enabling transfer. These are not soundness issues so much as opportunities for deeper explanation.

## Presentation
The presentation is excellent. The paper is very well organized, and each section builds naturally on the previous one. I found the preliminaries especially clear and useful to refer back to throughout the paper. The experiments, results, and discussion in Sections 4, 5, and 6 are also laid out in a very effective way, and the take-away summaries make the main conclusions easy to follow. The figures are similarly well designed: the plots are organized clearly, and the captions do a very good job of explaining how they should be interpreted. Overall, the paper is very clear, and I rarely felt there was room for confusion or ambiguity.

## Significance
I think this paper is quite significant for machine learning, especially for language model pretraining. The main result could have substantial practical impact: procedural pretraining may improve data efficiency and reduce the amount of semantic data needed to reach similar performance. More broadly, the work suggests a promising direction for thinking about how pretraining curricula can be structured more effectively. I also found the weight-mixture experiments especially interesting, since they open up possible future work not only on training efficiency but also on model interpretability and modularity.

## Originality
Although the paper is clearly inspired by prior work, including Hu et al. (2025) and Papadimitriou & Jurafsky (2023), I do not think that detracts from its originality. The paper contains a large number of new experiments, and the empirical scope is much broader than in prior work. In particular, the procedural-data mixture experiments and the weight-mixture experiments both stand out as creative and interesting additions. Overall, the paper feels novel not because it begins from a completely new idea, but because it pushes that idea much further and in several new directions.

---

> ### Author Rebuttal · Authors · 2026-03-30
>
> Many thanks for this very encouraging review. We are glad that the experiments were deemed comprehensive, and the results novel and important.
>
> ---
>
> **`Q1: Why particular procedural pretraining tasks transfer to particular downstream abilities?`**
>
> This is an important question indeed. We already have the following elements of answer.
> - Section 4 shows that each type of data improves specific computational capabilities.
> - Appendix F contains a series of experiments where we show that the effects of procedural pretraining cannot be replicated with low-level mechanistic interventions.
> F1 shows that the benefits do not arise from a generic sharpening of the attention.
> F2 shows that they do not arise simply from the distribution of parameter magnitudes.
> F2 also shows that precise patterns and weight values are essential, and coarse statistics alone cannot account for the observed benefits.
> - Section 4.1/Figure 2 show that precise structure *in the procedural data* is critical to its effects; using data with a similar token distribution but without the original structure fails to provide benefits.
>
> The effects are therefore complex to pinpoint and describe. We hypothesize for example that k-Dyck data improves on the "*Haystack*" diagnostic task because tracking balanced parentheses involves long-range dependencies, which is functionally similar to the key-value retrieval in *Haystack*. ECA-110 may help on "*Reversed Addition*" because the ECA involves local, deterministic state transitions that propagate information in one direction, which mirrors the left-to-right carry propagation in *Reversed Addition*.
> Verifying these hypotheses could be possible with the full toolset of mechanistic intepretability (mentioned in the discussion). This is a significant opportunity for future/deeper explanations.
>
> ---
>
> **`Q2: Under what conditions are transferred skills primarily stored in MLPs, and when is attention the main component enabling transfer?`**
>
> Sections 5.1-5.2 show that this depends on the target domain (what kind of data the model is ultimately trained on).
> - Natural language benefits more from skills stored in MLPs.
> - Structured domains (e.g. pure code) benefit more from skills stored in the attention.
> - For domains containing both (e.g. code with documentation, free-form math), attention and MLP are both useful.
>
> Therefore, procedural pretraining affects the MLPs and attention layers in distinct, complementary ways.
> In Appendix F, we report a series of experiments showing that the above benefits cannot be replicated with low-level interventions (sharpening the attention for example, or adjusting the distribution of magnitudes in MLP weights).
> This suggests that matching domain-specific and component-specific effects also requires more than low-level statistics of the data.
> This is definitely an area worth further exploration (together with Q1). We will highlight this in the discussion.
>
> ---
>
> Thanks again for the feedback!

---

> > ### Author Rebuttal · Reviewer_87jz · 2026-04-02
> >
> > Thank you to the authors for their response. I will maintain my positive score.

---

> > > ### Author Response · Authors · 2026-04-05
> > >
> > > Thank you for your continued engagement, and we are glad that our response addresses your questions!

---

### Official Review · Reviewer_eRck · 2026-03-12

**Soundness:** 1
**Presentation:** 2
**Significance:** 3
**Originality:** 3
**Overall Recommendation:** 4
**Confidence:** 3

**Summary:**

The authors use procedural "pre-pre-training" on LLMs. i.e. they start the pre-training procedure by training on some procedurally-generated task and show how it increases performance on certain downstream tasks. They show a connection between the chosen procedural data and the downstream tasks that are affected by it; They show some analysis on localization of the knowledge induced by procedural pre-pretraining;

**Compliance With Llm Reviewing Policy:**

Affirmed.

**Final Justification:**

My questions were addressed in the authors' rebuttal, but some concerns remain, therefore I overall recommend a weak accept.

**Key Questions For Authors:**

- How similar is the "standard data" (T2, line 133) used in this work to general pretraining data, like "The Pile" dataset and similar sized general-purpose text datasets? Don't the authors find this difference to be a critical difference, that might affect the validity and generalizability of their findings? (I saw that the authors used the Math part of The Pile in section 5.2, but not in other sections).
- In appendix C, the authors state they use a different-size architecture for each task. Why? This raises some concerns on the generalizability of the paper's findings (It makes sense that some tasks can't be solved with the smaller transformers, that's fine. But why not use larger-sized transformers (12 blocks) for all tested tasks?).

Minor suggestions and edits:

- I initially found many "procedural pretraining" works missing from the related work section. Having found they appear in the Extended literature review appendix, I encourage the authors to add a reference to that section from section 2.
- In line 153, right column, m_u shouldn't be part of the key-value list, or the writing should be rephrased.

**Limitations:**

Yes, the authors discuss limitations.

**Strengths And Weaknesses:**

Strengths:

- The presented approach of bootstrapping pretraining with procedural data, despite not being completely original (existed for quite some time in vision models), is looked at through a new lens in LLMs, and the paper shows interesting findings in that direction. I think this question is very interesting and has many possible further directions.
- The work not only suggests and evaluates certain settings of the procedural-pre-pre-training approach, but also investigates different wide questions around it - regarding localization, effect of different types of procedural data on downstream tasks, and analysis of savings of pretraining tokens when replacing them with procedural data.


Weaknesses:

I think there are two main weaknesses - one is lack of clarity of certain parts of the writing (exemplified below), and the second (more major one) is the use of different experimental settings (model size, standard training data choice) to answer different questions along this paper.

To elaborate:

Generally, my main problem with this paper is the usage of different settings across different experiments (see Question 2 below).
This doesn't only harm the validity of conclusions across the paper, but also makes the entire point weaker. i.e. - if the authors want to show procedural pre-pretraining can complement standard pre-training, they should choose a single model size (for all non-size-related experiments), a standard pretraining dataset (like The Pile, or anything else that falls within reasonable compute limitations), and evaluate with standard metrics against that dataset (and possibly a variety of downstream tasks).
This is in contrast to the current framing, in which the authors show procedural pre-pretraining can help on SOME (possibly cherry-picked?) downstream tasks (and these findings also depend on model size).

There are some unclear explanations. Some examples:
- the authors present (in line119) all non-procedural training data as "semantic data", but then reference to some non-procedural data being "semantic" and some "algorithmic" (line 134-135). Is this a typo and the title in line 119 should've been "standard data"?
- An unclear visualization: The bar colors in figure 2 should be switched to something more distinct, the bars should be in the same order across the evaluation plots AND the legend should be in the same order as the bar columns. Right now it's difficult to map 1-to-1 on the procedural data type to resulting bar. Right now this figure is unclear on its own.
- The localization scheme in section 4.2 deviates from standard approaches, without justification: Copying only specific sub-layers and **randomizing others** can cause unexpected effects and make the resulting evaluations un-trustworthy. Do the authors have any reference for a work that used this procedure to support their method? Otherwise, I would expect some validation on this method, or (even better) just use standard methods for localization like activation patching / attribution patching (attributing the correct answer logits in each prompt to each sub-layer's output), which don't take much compute / runtime.
- The method in section 5.1 is unclear. Which section of the pretraining datasets (WikiText and JavaCorpus) do the authors use as T2? All of the 15M and 105M tokens (respectively) or only some subset of them? This isn't mentioned. Also in this section, while the authors claim that procedural pre-pretraining helps across domains, it doesn't seem to lead to a significant improvement for Coding. This is brushed off and isn't addressed.

I would appreciate the authors addressing these concerns and answering the questions below to increase my score.

*Another note for the authors, which I don't hold as a weakness for this paper - Given that there is already quite some literature on procedural pre-pretraining, I think the most interesting part in this work is to expand section 4.3 (understanding WHY procedural data helps). I appreciate the authors writing section 4.3 and mentioning this in future work, but I think this is REALLY the interesting part, and its absence is felt.

---

> ### Author Rebuttal · Authors · 2026-03-30
>
> Many thanks for the thoughtful review. We are glad that the analysis and findings were deemed "*comprehensive*" and "*very interesting*", with "*many possible further directions*". We particularly appreciate the recognition of Section 4.3 (*why* the method works).
>
> **The main concern seems about the clarity of the experimental settings and explanations**. We address this below and are updating the manuscript.
>
> ---
>
> **`W1 & Q2: Different settings for different experiments`**
>
> We respectfully disagree that this is a weakness, but we understand the confusion and explain the rationale for these choices.
>
> Our experiments contain two main parts.
> 1. (Section 4) The first part identifies why/when procedural training helps, using **algorithmic tasks as diagnostic tools** .
> 2. (Section 5) The second part shows, with much larger models, that **procedural training improves pretraining in several domains** of practical interest across model/data scales .
>
> In the first part (algorithmic tasks), the model size is kept small (2 layers):
> - to avoid performance saturation through scale (except for multiplication, we use a 4-layer model to avoid underfitting),
> - to enable reliable conclusions by running each experiments with *10 different seeds*.
> These algorithmic tasks are **not** to display improvements. They are standard diagnostic tasks that probe the nature of the improvements (e.g. needle-in-a-haystack reflects long-context recall).
>
> In the second part, the model size ranges from 124M to 350M and 1.3B. We use standard pretraining datasets covering diverse domains (language/code/math). The **same procedural data as studied in the first part** shows consistent improvements.
>
> **Nothing is cherry-picked: the diagnostic algorithmic tasks are chosen diversely, the pretraining datasets are standard, and model sizes cover a wide range**. This directly addresses the generalizability concern.
>
> ---
>
> **`W2: Unclear explanations`**
>
> Thanks for pointing these out, we have updated the text to address the following points.
> - **Definitions of standard/semantic data**.
>
>   The reviewer is correct. We've now made the terminology more consistent. We define "*semantic data*" (standard pretraining data such as language, code, math) and "*algorithmic data*" (data for diagnostic purposes to probe basic capabilities, such as needle-in-a-haystack, addition, etc.). In our method, the model undergoes a procedural warm-up, then is exposed to either of these two data types, the former to show improvements on pretraining, and the latter as a diagnostic tool.
>
> - **Unclear visualization (Figure 2).** Well noted, we updated the color palette and re-ordered the legend as suggested.
>
> - **Localization scheme (partial transfer).**
>
>   The activation patching (suggested by the reviewer) addresses a very different question ("*which activations are causally responsible for a behavior*"). Instead, we identify which parts of the architecture benefit from a procedural warmup. Therefore, partial transfer is the most direct and obvious since **it is simply an ablation of the proposed method**.
>
>   This is widely used in transfer learning (i.e. initializing a model with a pretrained backbone while retraining top layers from scratch). And it was specifically proposed for LLMs in [1] to study transferrability of pretrained layers. See [2] for another recent example.
>
>   [1] Tamkin et al., *Investigating Transferability in Pretrained Language Models*, EMNLP 2020
>
>   [2] Fadlon et al. *How Much Pretraining Does Structured Data Need*, ACL 2026
>
> - **Method in Section 5.1.**
>     - *Dataset details*: The requested details are in Appendix E.3 (L847). We use all the data for each.
>     - *Significance of results on coding*: the perplexity ~8.8 → 8 is a substantial relative improvement (10%). The improvement varies by type of procedural data, which is itself an interesting finding. Indeed, we **further verify the improvement with much larger scales** for code (Figure 5). In response to **`Reviewer ci8x (W2)`**, we add **new downstream results on MBPP**, our model achieves non-random capability while the baseline **completely** fails.
>
> ---
>
> **`Q1. How similar is our data to general-purpose text datasets?`**
>
> We use standard data that is part of typical mixtures used to pretrain LLMs.
> - The Deepmind-Math dataset is a part of The Pile.
> - The language dataset (C4) is a filtered version of Common Crawl and a standard clean pretraining corpus, similar to the web part of The Pile.
> - The code dataset (CodeParrot) is a well-known collection from Github, which has been used by Huggingface as the sole data for training the 1.5B CodeParrot model, for example.
>
> The data we use is therefore representative of standard pretraining data.
> We prefer domain-specific datasets over heterogeneous mixtures like The Pile, so that we can examine the effects of our method on each domain.
>
> ---
>
> Thanks again, we hope our response addresses your concerns and look forward to your updated score!

---

> > ### Author Rebuttal · Reviewer_eRck · 2026-04-04
> >
> > I thank the authors for their detailed response.
> >
> > My questions were addressed, and I'm raising my score accordingly.
> > Additionally, regarding my main discussed weakness (W1), I would urge the authors to add the clarifications into the paper (either as some sort of discussion appendix or via clarifying these parts in the main paper).

---

> > > ### Author Response · Authors · 2026-04-05
> > >
> > > Thank you for your continued engagement, and we are glad that our response addresses your questions!
> > >
> > > We will add clarifications in the main paper regarding W1.

---

### Official Review · Reviewer_gWnn · 2026-03-13

**Soundness:** 3
**Presentation:** 4
**Significance:** 3
**Originality:** 3
**Overall Recommendation:** 5
**Confidence:** 2

**Summary:**

The paper studies training on procedural data as a means for LLMs to acquire rich semantic knowledge. The key idea is to reduce the dependency on web-scale data. The paper explores the impact of training on different types of procedural data on different types of (synthetic tasks). Guided by this exploration, the paper studies whether procedural training improves performance during pre-training and whether the gains persist after fine-tuning. The paper shows that procedural training can improve model performance with less pre-training data across language, code, and math.

**Compliance With Llm Reviewing Policy:**

Affirmed.

**Final Justification:**

The authors have addressed the concerns raised in the review. I maintain my original score (5).

**Key Questions For Authors:**

1. In Section 5.1, why did the authors perform full model transfer when, in Section 4.2, they claim transferring specific components can be better?
2. Why does training on procedural data help language significantly more than on code and math?
3. What is the number of procedural tokens used for full-model, MLP and attention transfer in Section 5.3?
4. Which model was fine-tuned on each benchmark – which weights were transferred between procedural and pretraining?

**Limitations:**

Yes

**Strengths And Weaknesses:**

**Strengths:**
1. The authors performed a detailed analysis of how procedural data impacts performance on different types of tasks.
2. The authors show that training on procedural data is complementary to standard pre-training and leads to data-efficient pre-training. This is a very significant finding given the persistent need for additional data for training transformers, and it can lead to energy and cost-effective training.
3. Generalizable solution: The proposed methodology can be used for multiple domains (language, code, and maths).
4. The paper is well written and easy to follow.

**Weaknesses:**
1. Some claims/conclusions are not accurate. For example, the authors claim that “transferring specific components can be more effective than transferring the entire model.” (Line 187, right). However, from Figure 3, transferring the entire model leads to overall better performance. Line 367 (left), the claim that transferring full-model is most effective on WiKiText does not hold – transferring full model has the most impact. Similarly, on CodeParrot MLP transfer has the best performance with union data.
2. The authors do not perform an independent analysis on the impact of different procedural data generation methods on math. They use the observation from code (union, set) for math. They also do not study the impact of fine-tuning on any math benchmark.
3. Weak validation on fine-tuning. The study on code is performed on a very old dataset (Py150), and the performance on GLUE does not show significant improvement on most tasks.

---

> ### Author Rebuttal · Authors · 2026-03-30
>
> Many thanks for the thoughtful review. We appreciate the recognition of our "*detailed analysis*" and "*very significant finding*".
>
> **Most concerns seem about the clarity of the writing and choices of benchmarks.** We provide clarifications/new results below that we are adding to the manuscript. We hope this will strengthen your confidence in our contributions.
>
> ---
>
> **`W1: Clarity of claims on partial transfer`**
>
> Our phrasing was indeed imprecise. Partial transfer is **not** universally better. Rather, it **can be superior in certain settings**. For example in Figure 3, needle-in-a-haystack benefits more from attention-only transfer **for some types of procedural data**, while MLP-only transfer is always bad. In comparison, for reversed-addition, MLP-only transfer is better. Similar for pretraining scenarios in Figure 6. This is an interesting finding: the localization of useful/transferrable information **varies across tasks/domains**, and transferring irrelevant components can be detrimental.
>
> The confusion on WikiText is due to an editing typo (**more → most**), our apologies. Transferring MLPs for WikiText is **more** effective than attention (and the opposite is true for JavaCorpus). The caption of Figure 6 conveys the correct message.
>
> ---
>
> **`W2: Impact of different types of procedural data studied on code, but not math`**
>
> On the Deepmind-Math dataset, we used the top performers identified on language and code (Section 5.1). Our goal was to **delibrately test whether these generalize** to the math domain.
> We propose to perform a full sweep over data types for DeepMind-Math and include the results in the final version of the paper.
>
> We are also actively looking into adding an external math benchmark (e.g. GSM8K) in the next revision of the paper.
>
> ---
>
> **`W3: Additional results on downstream benchmarks`**
>
> We initially chose PY150 (code-completion) and GLUE (NLU tasks) because they are well-suited and standard for the model scale considered. The improvement on PY150 (60.5 → 62.1) is meaningful. On GLUE, the average improves 68.5 → 71.3, with gains of +10 points on certain tasks.
>
> To strenghten our results, we performed **new additional experiments** of our code and language models on coding (MBPP) and general reasoning (ARC-Easy, HellaSwag) benchmarks, reported in the response to **`Reviewer ci8x (W2)`**.
> Our method brings significant improvements using only a tiny amount of procedural data.
>
> ---
>
> **`Q1: Why not use partial transfer in large experiments?`**
>
> Partial transfer is indeed better is some cases, but it's quite task-specific (see W1). We use full transfer in all experiments (except for Sections  4.2 and 5.3) since it is simply **more aligned with common practices, performs well on average, and is practically straightforward to deploy**.
>
> The partial-transfer experiments (Sections 4.2 and 5.3) are primarily aimed at **understanding the mechanisms**  behind the observed improvements (localization of useful information).
>
> ---
>
> **`Q2: Why does procedural pretraining help language significantly more than code/math?`**
>
> First, in terms of loss/accuracy improvements, procedural pretraining helps code/math as much as language:
> - Language (C4): perplexity ~60 → 56 (7% relative improvement),
> - Code (CodeParrot): perplexity ~5.4 → 5.1 (6% relative improvement),
> - Math (DeepMind-Math): problem accuracy (which is harder/more rigorous) 40.5% → ~43% (a substantial absolute accuracy improvement).
>
> Second, in terms of token savings, procedural pretraining helps language more than code/math (saving more semantic tokens to achieve a given performance). We hypothesize that this is because code and math data are already highly structured and contain some of the structural regularities that make procedural data useful, leaving less room for improvement.
>
> ---
>
> **`Q3: Number of procedural tokens used in Section 5.3?`**
>
> Same as Section 5.2: 2-10M. The only difference between Sections 5.2 and 5.3 is the *partial* transfer.
>
> ---
>
> **`Q4: Which model was fine-tuned on each benchmark – which weights were transferred between procedural and pretraining?`**
>
> The best models in Figure 5 (SORT for C4, UNION for CodeParrot) are evaluated on downstream benchmarks.
> This is stated in Appendix M but we will make it more prominent.
> Full transfer is used between procedural and semantic pretraining.
>
> ---
> Thanks again for the feedback. We are updating the paper accordingly, and let us know if there is any remaining question!

---

> > ### Author Rebuttal · Reviewer_gWnn · 2026-04-02
> >
> > Thank you for the response and clarification.
> >
> > From what I understand (response to W2 and Q4), the findings from code can generalize to math but these do not generalize between language and code. Is this correct? A small discussion around generalization of findings to new domains (what works and what does not) might be beneficial for future research in this direction.
> >
> > I will keep the positive score.

---

> > > ### Author Response · Authors · 2026-04-02
> > >
> > > Thank you for your continued engagement!
> > >
> > > Regarding your comment, there are certain types of data that generalize well across all three domains (e.g., UNION, see Figure 5 top), but the best performers for language and code/math are indeed different (SORT and UNION, respectively).
> > >
> > > We will add a short discussion on this in the next revision.

---

### Decision · Program_Chairs · 2026-04-30

**Decision:**

Accept (spotlight)

**Comment:**

This paper shows that a short warm-up on synthetic procedural data before pretraining can accelerate convergence and improve downstream results in various reasoning & context recall areas areas. The work uses strong controls like shuffles to isolate structure from superficial statistics. There were some concerns about claims and experiments initially, but many of these were addressed in the rebuttal phase, and the majority of the reviewers and I agree that this is a strong paper that will be valuable to the field.